# PRIVACY IMPLICATIONS OF SHUFFLING

**Casey Meehan** [1]**, Amrita Roy-Chowdhury** [2]**, Kamalika Chaudhuri** [1]**, Somesh Jha** [2]
[1]UC San Diego,  [2] University of Wisconsin, Madison

## ABSTRACT

LDP deployments are vulnerable to inference attacks as an adversary can link the noisy responses to their identity and subsequently, auxiliary information using the *order* of the data. An alternative model, shuffle DP, prevents this by shuffling the noisy responses uniformly at random. However, this limits the data learnability – only symmetric functions (input order agnostic) can be learned. In this paper, we strike a balance and show that systematic shuffling of the noisy responses can thwart specific inference attacks while retaining some meaningful data learnability. To this end, we propose a novel privacy guarantee, $d_\sigma$-privacy, that captures the privacy of the order of a data sequence. $d_\sigma$-privacy allows tuning the granularity at which the ordinal information is maintained, which formalizes the degree the resistance to inference attacks trading it off with data learnability. Additionally, we propose a novel shuffling mechanism that can achieve $d_\sigma$-privacy and demonstrate the practicality of our mechanism via evaluation on real-world datasets.

## 1 INTRODUCTION

Differential Privacy (DP) and its local variant (LDP) are the most commonly accepted notions of data privacy. LDP has the significant advantage of not requiring a trusted centralized aggregator, and has become a popular model for commercial deployments, such as those of Microsoft (Ding et al., 2017), Apple (Greenberg, 2016), and Google (Erlingsson et al., 2014; Fanti et al., 2015; Bittau et al., 2017b). Its formal guarantee asserts that an adversary cannot infer the value of an individual's private input by observing the noisy output. However in practice, a vast amount of *public auxiliary information*, such as address, social media connections, court records, property records, income and birth dates (La Corte, 2019), is available for every individual. An adversary, with access to such auxiliary information, *can* learn about an individual's private data from several *other* participants' noisy responses. We illustrate this as follows.

> **Problem.** An analyst runs a medical survey in Alice's community to investigate how the prevalence of a highly contagious disease changes from neighborhood to neighborhood. Community members report a binary value indicating whether they have the disease.

Next, consider the following two data reporting strategies.

> **Strategy 1.** Each data owner passes their data through an appropriate randomizer (that flips the input bit with some probability) in their local devices and reports the noisy output to the untrusted data analyst.

> **Strategy 2.** The noisy responses from the local devices of each of the data owners are collected by an intermediary trusted shuffler which dissociates the device IDs (metadata) from the responses and uniformly randomly shuffles them before sending them to the analyst.

**Strategy 1** corresponds to the standard LDP deployment model (for example, Apple and Microsoft's deployments). Here *the order of the noisy responses is informative of the identity of the data owners* – the noisy response at index 1 corresponds to the first data owner and so on. Thus, the noisy responses can be directly linked with its associated device/account ID and subsequently, auxiliary information. This puts Alice's data under the threat of inference attacks. For instance, an adversary[1] may know the home addresses of the participants and use this to identify the responses of all the individuals from Alice's household. Being highly infectious, all or most of them will have the same true value (0 or 1). Hence, the adversary can reliably infer Alice's value by taking a simple majority vote of her and her household's noisy responses. Note that this does not violate the LDP guarantee since the inputs are appropriately randomized when observed in isolation. Additionally, on account of being

---

[1]The analyst and the adversary could be same, we refer to them separately for the ease of understanding.

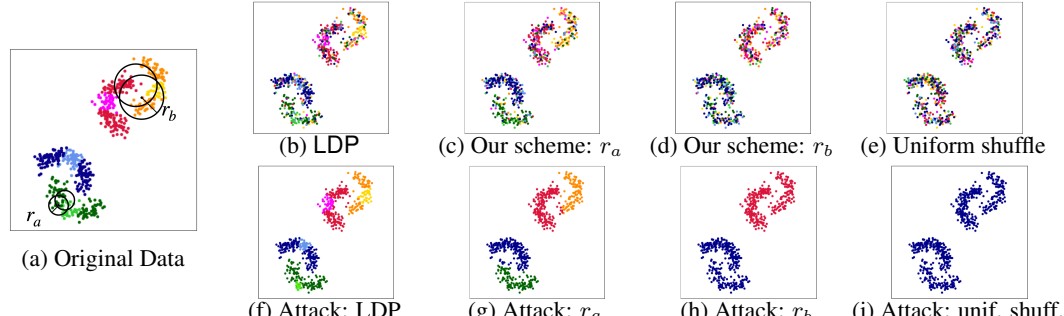

Figure 1: Demonstration of how our proposed scheme thwarts inference attacks at different granularities. Fig. 1a depicts the original sensitive data (such as income bracket) with eight color-coded labels. The position of the points represents public information (such as home address) used to correlate them. There are three levels of granularity: warm vs. cool clusters, blue vs. green and red vs. orange crescents, and light vs. dark within each crescent. Fig. 1b depicts $\epsilon = 2.55$ LDP. Fig. 1c and 1d correspond to our scheme, each with $\alpha = 1$ (privacy parameter, Def. 4.3). The former uses a smaller distance threshold ($r_1$, used to delineate the granularity of grouping – see Sec. 4.2) that mostly shuffles in each crescent. The latter uses a larger distance threshold ($r_2$) that shuffles within each cluster. Figures in the bottom row demonstrate an inference attack (uses Gaussian process correlation) on all four cases. We see that LDP reveals almost the entire dataset (Fig. 1f) while uniform shuffling prevents all classification (1i). However, the granularity can be controlled with our scheme (Figs. 1g, 1h).

public, the auxiliary information is known to the adversary (and analyst) *a priori* – no mechanism can prevent their disclosure. For instance, any attempts to include Alice's address as an additional feature of the data and then report via LDP is *futile* – the adversary would simply discard the reported noisy address and use the auxiliary information about the exact addresses to identify the responses of her household members. We call such threats *inference attacks* – recovering an individual's private input using all or a subset of other participants' noisy responses. It is well known that protecting against inference attacks that rely on underlying data correlations is beyond the purview of DP (Kifer & Machanavajjhala, 2014; Tschantz et al., 2020).

**Strategy 2** corresponds to the recently introduced shuffle DP model, such as Google's Prochlo (Bittau et al., 2017b). Here, the noisy responses are completely anonymized – the adversary cannot identify which LDP responses correspond to Alice and her household. Under such a model, only information that is completely order agnostic (i.e., symmetric functions that can be computed over just the *bag* of values, such as aggregate statistics) can be extracted. Consequently, the analyst also fails to accomplish their original goal as all the underlying data correlation is destroyed.

Thus, we see that the two models of deployment for LDP present a trade-off between vulnerability to inference attacks and scope of data learnability. In fact, as demonstrated in Kifer & Machanavajjhala (2011), it is impossible to defend against *all* inference attacks while simultaneously maintaining utility for learning. In the extreme case that the adversary knows *everyone* in Alice's community has the same true value (but not which one), no mechanism can prevent revelation of Alice's datapoint short of destroying all utility of the dataset. This then begs the question: ***Can we formally suppress specific inference attacks targeting each data owner while maintaining some meaningful learnability of the private data?*** Referring back to our example, can we thwart attacks inferring Alice's data using specifically her households' responses and still allow the medical analyst to learn its target trends? Can we offer this to every data owner participating?

In this paper, we strike a balance and propose a generalized shuffle framework that meets the utility requirements of the above analyst while formally protecting data owners against inference attacks. Our solution is based on the key insight: *the order of the data acts as the proxy for the identity of data owners* as illustrated above. The granularity at which the ordering is maintained formalizes resistance to inference attacks while retaining some meaningful learnability of the private data. Specifically, we guarantee each data owner that their data is shuffled together with a carefully chosen group of other data owners. Revisiting our example, consider uniformly shuffling the responses from Alice's household and her immediate neighbors. Now an adversary cannot use her household's responses to predict her value any better than they could with a random sample of responses from this group. In the same way that LDP prevents reconstruction of her datapoint using specifically *her* noisy response, this scheme prevents reconstruction of her datapoint using specifically *her households'* responses. The real challenge is offering such guarantees *equally* to *every* data owner. Bob, Alice's

neighbor, needs his households' responses shuffled in with his neighbors, as does Luis who is a neighbor of Bob but *not* of Alice. Thus, we have $n$ data owners with $n$ distinct, overlapping groups. Our scheme supports arbitrary groupings (overlapping or not), introducing a diverse and tunable class of privacy/utility trade-offs which is not attainable with either LDP or uniform shuffling alone. For the above example, our scheme can formally protect each data owner from inference attacks using specifically their household, while still learning how disease prevalence changes across the neighborhoods of Alice's community.

This work offers two key contributions to the machine learning privacy literature:

- **Novel privacy guarantee.** We propose a novel privacy definition, $d_\sigma$-privacy that captures the privacy of the *order* of a data sequence (Sec. 4.2) and formalizes the degree of resistance against inference attacks (Sec. 4.3). $d_\sigma$-privacy allows assigning an arbitrary group, $G_i$, to each data owner, $\mathsf{DO}_i, i \in [n]$. For instance, the groups can represent individuals in the same age bracket, 'friends' on social media, or individuals living in each other's vicinity (as in case of Alice in our example). Recall that the order is informative of the data owner's identity. Intuitively, $d_\sigma$-privacy protects $\mathsf{DO}_i$ from inference attacks that arise from knowing the *identity* of the members of their group $G_i$ (Sec. 4.3). Additionally, this grouping determines a threshold of learnability – any learning that is order agnostic within a group (disease prevalence in a neighborhood – the data analyst's goal in our example) is utilitarian and allowed; whereas analysis that involves identifying the values of individuals within a group (disease prevalence within specific households – the adversary's goal) is regarded as a privacy threat and protected against. See Fig. 1 for a toy demonstration of how our guarantee allows *tuning the granularity at which trends can be learned*.

- **Novel shuffle framework.** We propose a novel mechanism that shuffles the data systematically and achieves $d_\sigma$-privacy. This provides a generalized shuffle framework that interpolates between no shuffling (LDP) and uniform random shuffling (shuffle model) in terms of protection against inference attacks and data learnability.

## 2 RELATED WORK

The shuffle model of DP (Bittau et al., 2017a; Cheu et al., 2019; Erlingsson et al., 2019) differs from our scheme as follows. These works (1) study DP benefits of shuffling whereas we study the inferential privacy benefits, and (2) only study uniformly random shuffling where ours generalizes this to tunable, non-uniform shuffling (see App. A.15).

A steady line of work has studied inferential privacy (Kasiviswanathan & Smith, 2014; Kifer & Machanavajjhala, 2011; Ghosh & Kleinberg, 2016; Dalenius, 1977; Dwork & Naor, 2010; Tschantz et al., 2020). Our work departs from those in that we focus on *local* inferential privacy and do so via the new angle of shuffling.

Older works such as $k$-anonymity (Sweeney, 2002), $l$-diversity Machanavajjhala et al. (2007), Anatomy (Xiao & Tao, 2006) and others (Wong et al., 2010; Tassa et al., 2012; Xue et al., 2012; Choromanski et al., 2013; Doka et al., 2015) have studied the privacy risk of non-sensitive auxiliary information or 'quasi identifiers'. These works (1) focus on the setting of dataset release, whereas we focus on dataset collection, and (2) do not offer each data owner formal inferential guarantees, whereas we do. The De Finetti attack (Kifer, 2009) shows how shuffling schemes are vulnerable to inference attacks that correlate records to recover the original permutation of sensitive attributes. A strict instance of our privacy guarantee can thwart such attacks (at the cost of no utility, App. A.3).

## 3 BACKGROUND

**Notations. Boldface** (such as $\mathbf{x} = \langle x_1, \cdots, x_n \rangle$) denotes a data sequence (ordered list); normal font (such as $x_1$) denotes individual values and $\{\cdot\}$ represents a multiset or bag of values.

### 3.1 LOCAL DIFFERENTIAL PRIVACY

The local model consists of a set of data owners and an untrusted data aggregator (analyst); each individual perturbs their data using a LDP algorithm (randomizers) and sends it to the analyst. The LDP guarantee is formally defined as

**Definition 3.1.** [Local Differential Privacy, LDP Warner (1965); Evfimievski et al. (2003); Kasiviswanathan et al. (2008)] A randomized algorithm $\mathcal{M} : \mathcal{X} \to \mathcal{Y}$ is $\epsilon$-locally differentially private (or $\epsilon$-LDP ), if for any pair of private values $x, x' \in \mathcal{X}$ and any subset of output,

$$\Pr\big[\mathcal{M}(x) \in \mathcal{W}\big] \le e^\epsilon \cdot \Pr\big[\mathcal{M}(x') \in \mathcal{W}\big] \tag{1}$$

The shuffle model is an extension of the local model where the data owners first randomize their inputs. Additionally, an intermediate trusted shuffler applies a *uniformly random permutation* to

all the noisy responses before the analyst can view them. The anonymity provided by the shuffler requires less noise than the local model for achieving the same privacy.

## 3.2 MALLOWS MODEL

A permutation of a set $S$ is a bijection $S \mapsto S$. The set of permutations of $[n], n \in \mathbb{N}$ forms a symmetric group $S_n$. As a shorthand, we use $\sigma(\mathbf{x})$ to denote applying permutation $\sigma \in S_n$ to a data sequence $\mathbf{x}$ of length $n$. Additionally, $\sigma(i), i \in [n], \sigma \in S_n$ denotes the value at index $i$ in $\sigma$ and $\sigma^{-1}$ denotes its inverse. For example, if $\sigma = (1\ 3\ 5\ 4\ 2)$ and $\mathbf{x} = \langle 21, 33, 45, 65, 67 \rangle$, then $\sigma(\mathbf{x}) = \langle 21, 45, 67, 65, 33 \rangle$, $\sigma(2) = 3, \sigma(3) = 5$ and $\sigma^{-1} = (1\ 5\ 2\ 4\ 3)$.

Mallows model is a popular probabilistic model for permutations (MALLOWS, 1957). The mode of the distribution is given by the reference permutation $\sigma_0$ – the probability of a permutation increases as we move 'closer' to $\sigma_0$ as measured by rank distance metrics, such as the Kendall's tau distance (Def. A.2). The dispersion parameter $\theta$ controls how fast this increase happens.

**Definition 3.2.** For a dispersion parameter $\theta$, a reference permutation $\sigma_o \in S_n$, and a rank distance measure $\boldsymbol{\delta} : S_n \times S_n \mapsto \mathbb{R}$, $\mathbb{P}_{\Theta,\boldsymbol{\delta}}(\sigma : \sigma_0) = \frac{1}{\psi(\theta,\boldsymbol{\delta})} e^{-\theta \boldsymbol{\delta}(\sigma,\sigma_0)}$ is the Mallows model where $\psi(\theta,\boldsymbol{\delta}) = \sum_{\sigma \in S_n} e^{-\theta \boldsymbol{\delta}(\sigma,\sigma_0)}$ is a normalization term and $\sigma \in S_n$.

# 4 DATA PRIVACY AND SHUFFLING

In this section, we present $d_\sigma$-privacy and a shuffling mechanism capable of achieving the $d_\sigma$-privacy guarantee.

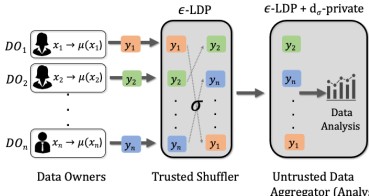

Figure 2: Trusted shuffler mediates on $\mathbf{y}$

## 4.1 PROBLEM SETTING

In our problem setting, we have $n$ data owners $\mathsf{DO}_i, i \in [n]$ each with a private input $x_i \in \mathcal{X}$ (Fig. 2). The data owners first randomize their inputs via a $\epsilon$-LDP mechanism to generate $y_i = \mathcal{M}(x_i)$. Additionally, just like in the shuffle model, we have a trusted shuffler. It mediates upon the noisy responses $\mathbf{y} = \langle y_1, \cdots, y_n \rangle$ to obtain the final output sequence $\mathbf{z} = \mathcal{A}(\mathbf{y})$ ($\mathcal{A}$ corresponds to Alg. 1) which is sent to the untrusted data analyst. The shuffler can be implemented via trusted execution environments (TEE) just like Google's Prochlo. Next, we formally discuss the notion of order and its implications.

**Definition 4.1.** (Order) The order of a sequence $\mathbf{x} = \langle x_1, \cdots, x_n \rangle$ refers to the indices of its set of values $\{x_i\}$ and is represented by permutations from $S_n$.

When the noisy response sequence $\mathbf{y} = \langle y_1, \cdots, y_n \rangle$ is represented by the identity permutation $\sigma_I = (1\ 2\ \cdots\ n)$, the value at index 1 corresponds to $\mathsf{DO}_1$ and so on. Standard LDP releases the identity permutation w.p. 1. The output of the shuffler, $\mathbf{z}$, is some permutation of the sequence $\mathbf{y}$, i.e.,
$$\mathbf{z} = \sigma(\mathbf{y}) = \langle y_{\sigma(1)}, \cdots, y_{\sigma(n)} \rangle$$
where $\sigma$ is determined via $\mathcal{A}(\cdot)$. For example, for $\sigma = (4\ 5\ 2\ 3\ 1)$, we have $\mathbf{z} = \langle y_4, y_5, y_2, y_3, y_1 \rangle$ which means that the value at index 1 ($\mathsf{DO}_1$) now corresponds to that of $\mathsf{DO}_4$ and so on.

## 4.2 DEFINITION OF $d_\sigma$-PRIVACY

Inferential risk captures the threat of an adversary who infers $\mathsf{DO}_i$'s private $x_i$ using all or a subset of other data owners' released $y_j$'s. Since we cannot prevent all such attacks and maintain utility, our aim is to formally limit *which data owners* can be leveraged in inferring $\mathsf{DO}_i$'s private $x_i$. To make this precise, each $\mathsf{DO}_i$ may choose a corresponding group, $G_i \subseteq [n]$, of data owners.

$d_\sigma$-privacy guarantees that $y_j$ values originating from a data owner's group $G_i$ are shuffled together. In doing so, the LDP values corresponding to subsets of $\mathsf{DO}_i$'s group $I \subset G_i$ cannot be reliably identified, and thus cannot be singled out to make inferences about $\mathsf{DO}_i$'s $x_i$. If Alice's group includes her whole neighborhood, LDP data originating from her household cannot be singled out to recover her private $x_i$.

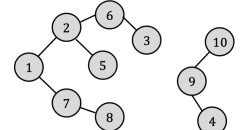

Figure 3: An example social media connectivity graph $\mathbf{t}_{e.g}$

Any choice of grouping $\mathcal{G} = \{G_1, G_2, \ldots, G_n\}$ can be accommodated under $d_\sigma$-privacy. Each data owner may choose a group large enough to hide anyone they feel sufficient risk from. We outline two systematic approaches to assigning groups as follows:

- Let $\mathbf{t} = \langle t_1, \cdots, t_n \rangle, t_i \in \mathcal{T}$ denote some public auxiliary information about each individual. $\mathsf{DO}_i$'s group, $G_i$, could consist of all those $\mathsf{DO}_j$'s who are similar to $\mathsf{DO}_i$ w.r.t. the public auxiliary information $t_i, t_j$ according to some distance measure $d : \mathcal{T} \times \mathcal{T} \to \mathbb{R}$. Here, we define

'similar' as being under a threshold[2] $r \in \mathbb{R}$ such that $G_i = \{j \in [n] | d(t_i, t_j) \leq r\}, \forall i \in [n]$. For example, $d(\cdot)$ can be Euclidean distance if $\mathcal{T}$ corresponds to geographical locations, thwarting inference attacks leveraging one's household or immediate neighbors. If $\mathcal{T}$ represents a social media connectivity graph, $d(\cdot)$ can measure the path length between two nodes, thwarting inference attacks using specifically one's close friends. For the example social media connectivity graph depicted in Fig. 3, assuming distance metric path length and $r = 2$, the groups are defined as $G_1 = \{1, 7, 8, 2, 5, 6\}, G_2 = \{2, 1, 7, 5, 6, 3\}$ and so on.

- Alternatively, the data owners might opt for a group of a specific size $r < n$. Collecting private data from a social media network, we may set $r = 50$, where each $G_i$ is encouraged to include the 50 data owners $\mathsf{DO}_i$ interacts with most frequently.

Intuitively, $d_\sigma$-privacy protects $\mathsf{DO}_i$ against inference attacks that leverages correlations at a finer granularity than $G_i$. In other words, under $d_\sigma$-privacy, one subset of $k$ data owners $\subset G_i$ (e.g. household) is no more useful for targeting $x_i$ than any other subset of $k$ data owners $\subset G_i$ (e.g. some combination of neighbors). This leads to the following key insight for the formal privacy definition.

**Key Insight.** Formally, our privacy goal is to prevent the leakage of ordinal information from within a group. We achieve this by systematically *bounding the dependence of the mechanism's output on the relative ordering (of data values corresponding to the data owners) within each group*.
First, we introduce the notion of neighboring permutations.

**Definition 4.2.** (Neighboring Permutations) Given a group assignment $\mathcal{G}$, two permutations $\sigma, \sigma' \in \mathrm{S}_n$ are defined to be neighboring w.r.t. a group $G_i \in \mathcal{G}$ (denoted as $\sigma \approx_{G_i} \sigma'$) if $\sigma(j) = \sigma'(j) \; \forall j \notin G_i$.

Neighboring permutations differ only in the indices of its corresponding group $G_i$. For example, $\sigma = (\underline{1}\,\underline{2}\,4\,5\,\underline{7}\,\underline{6}\,\underline{10}\,3\,8\,9)$ and $\sigma' = (\underline{7}\,3\,4\,5\,\underline{6}\,\underline{2}\,\underline{1}\,\underline{10}\,8\,9)$ are neighboring w.r.t $G_1$ (Fig. 3) since they differ only in $\sigma(1), \sigma(2), \sigma(5), \sigma(6), \sigma(7)$ and $\sigma(8)$. We denote the set of all neighboring permutations as

$$\mathrm{N}_\mathcal{G} = \{(\sigma, \sigma') | \sigma \approx_{G_i} \sigma', \exists G_i \in \mathcal{G}\} \tag{2}$$

Now, we formally define $d_\sigma$-privacy as follows.

**Definition 4.3** ($d_\sigma$-privacy). For a given group assignment $\mathcal{G}$ on a set of $n$ entities and a privacy parameter $\alpha \in \mathbb{R}_{\geq 0}$, a randomized mechanism $\mathcal{A} : \mathcal{Y}^n \mapsto \mathcal{V}$ is $(\alpha, \mathcal{G})$-$d_\sigma$ private if for all $\mathbf{y} \in \mathcal{Y}^n$ and neighboring permutations $\sigma, \sigma' \in \mathrm{N}_\mathcal{G}$ and any subset of output $O \subseteq \mathcal{V}$, we have

$$\Pr[\mathcal{A}(\sigma(\mathbf{y})) \in O] \leq e^\alpha \cdot \Pr[\mathcal{A}(\sigma'(\mathbf{y})) \in O] \tag{3}$$

$\sigma(\mathbf{y})$ and $\sigma'(\mathbf{y})$ are defined to be *neighboring sequences*.

$d_\sigma$-privacy states that, for any group $G_i$, the mechanism is (almost) agnostic of the order of the data within the group. Even after observing the output, an adversary cannot learn about the relative ordering of the data within any group. Thus, two neighboring sequences are indistinguishable to an adversary. An important property of $d_\sigma$-privacy is that post-processing computations does not degrade privacy. Additionally, when applied multiple times, the privacy guarantee degrades gracefully. Both the properties are analogous to $\mathsf{DP}$ and are presented in App. A.4.

**Note.** Any data sequence $\mathbf{x} = \langle x_1, \cdots, x_n \rangle$ can be viewed as a two-tuple, $(\{x\}, \sigma)$, where $\{x\}$ denotes the *bag* of values and $\sigma \in S_n$ denotes the corresponding indices of the values which represents the *order* of the data. The $\epsilon$-LDP protects the bag of data values, $\{x\}$, while $d_\sigma$-privacy protects the order, $\sigma$. Thus, the two privacy guarantees cater to orthogonal parts of a data sequence (see Thm. 4.2 ). Also, $\alpha = \infty\ (0), r = 0\ (n)$ represents the standard $\mathsf{LDP}$ (shuffle $\mathsf{DP}$) setting.

### 4.3 Privacy Implications

The group assignment $\mathcal{G}$ delineates a threshold of learnability which determines the privacy/utility tradeoff as follows.

- **Learning allowed (Analyst's goal)**. $d_\sigma$-privacy can answer queries that are order agnostic within groups, such as aggregate statistics of a group. In Alice's case, the analyst can estimate the disease prevalence in her neighborhood.

- **Learning disallowed (Adversary's goal)**. Adversaries cannot identify (noisy) values of individuals within any group. While they may learn the disease prevalence in Alice's neighborhood, they cannot determine the prevalence within her household and use that to recover her value $x_i$.

To make this precise, we first formalize the privacy implications of the $d_\sigma$ guarantee in the standard Bayesian framework, typically used for studying inferential privacy. Next, we formalize the privacy provided by the combination of $\mathsf{LDP}$ and $d_\sigma$ guarantees by way of a decision theoretic adversary.

---

[2]We could also have different thresholds, $r_i$, for every data owner, $\mathsf{DO}_i$.

**Bayesian Adversary.** Consider a Bayesian adversary with any prior $\mathcal{P}$ on the joint distribution of noisy responses, $\Pr_{\mathcal{P}}[\mathbf{y}]$, which models their beliefs on the correlation between the participants (such as the correlation between Alice and her households' disease status). Their goal is to infer $\mathsf{DO}_i$'s private input $x_i$. As with early $\mathsf{DP}$ works (Dwork et al., 2006), we consider an *informed* adversary. Here, the adversary knows (1) the sequence (assignment) of noisy values outside $G_i$, $\mathbf{y}_{\overline{G}_i}$, and (2) the (unordered) bag of noisy values in $G_i$, $\{y_{G_i}\}$. $d_\sigma$-privacy bounds the prior-posterior odds gap on $x_i$ for such as informed adversary as follows:

**Theorem 4.1.** *For a given group assignment $\mathcal{G}$ on a set of $n$ data owners, if a shuffling mechanism $\mathcal{A} : \mathcal{Y}^n \mapsto \mathcal{Y}^n$ is $(\alpha, \mathcal{G})$-$d_\sigma$ private, then for each data owner $\mathsf{DO}_i, i \in [n]$,*

$$\max_{\substack{i \in [n] \\ a, b \in \mathcal{X}}} \left| \log \frac{\Pr_{\mathcal{P}}[x_i = a | \mathbf{z}, \{y_{G_i}\}, \mathbf{y}_{\overline{G}_i}]}{\Pr_{\mathcal{P}}[x_i = b | \mathbf{z}, \{y_{G_i}\}, \mathbf{y}_{\overline{G}_i}]} - \log \frac{\Pr_{\mathcal{P}}[x_i = a | \{y_{G_i}\}, \mathbf{y}_{\overline{G}_i}]}{\Pr_{\mathcal{P}}[x_i = b | \{y_{G_i}\}, \mathbf{y}_{\overline{G}_i}]} \right| \leq \alpha$$

*for a prior distribution $\mathcal{P}$, where $\mathbf{z} = \mathcal{A}(\mathbf{y})$ and $\mathbf{y}_{\overline{G}_i}$ is the noisy sequence for data owners outside $G_i$.* See App A.5 for the proof and further discussion on the semantic meaning of the above guarantee.

**Decision Theoretic Adversary.** Here, we analyse the privacy provided by the combination of $\mathsf{LDP}$ and $d_\sigma$ guarantees. Consider a decision theoretic adversary who aims to identify the noisy responses, $\{z_I\}$, that originated from a specific subset of data owners, $I \subset G_i$ (such as the members of Alice's household). We denote the adversary by a (possibly randomized) function mapping from the output $\mathbf{z}$ sequence to a set of $k$ indices, $\mathcal{D}_{Adv} : \mathcal{Y}^n \to [n]^k$, where $k = |I|$. These $k$ indices, $H \in [n]^k$, represent the elements of $\mathbf{z}$ that $\mathcal{D}_{Adv}$ believes originated from the data owners in $I$. $\mathcal{D}_{Adv}$ wins if $> k/2$ of the chosen indices indeed originated from $I$, i.e, $|\sigma(H) \cap I| > k/2$, where $z_i = y_{\sigma(i)}$ and $\sigma(H) = \{\sigma(i) : i \in H\}$. $\mathcal{D}_{Adv}$ loses if most of $H$ did not originate from $I$, i.e., $|\sigma(H) \cap I| \leq k/2$. We choose the above adversary because this re-identification is a key step in carrying out inference attacks – in failing to reliably re-identify the noisy values originating from $I$, one cannot make inferences on $x_i$ specifically from the subset $I \subset G_i$.

**Theorem 4.2.** *For $\mathcal{A}(\mathcal{M}(\mathbf{x})) = \mathbf{z}$ where $\mathcal{M}(\cdot)$ is $\epsilon$-$\mathsf{LDP}$ and $\mathcal{A}(\cdot)$ is $\alpha$ - $d_\sigma$ private, we have*

$$\Pr[\mathcal{D}_{Adv} \text{ loses}] \geq \left\lfloor \frac{r - k}{k} \right\rfloor e^{-(2k\epsilon + \alpha)} \cdot \Pr[\mathcal{D}_{Adv} \text{ wins}]$$

*for any input subgroup $I \subset G_i, r = |G_i|$ and $k < r/2$.*

The adversary's ability to re-identify the $\{z_I\}$ values comes partially from the *bag of values* (quantified by $\epsilon$) and partially from the *order* (quantified by $\alpha$). We highlight two implications of this fact.

- When $\epsilon$ is small ($\ll 1$), an adversary's ability to re-identify the noisy values $\{z_I\}$ originating from $I$ may very well be dominated by $\alpha$. For instance, if $\epsilon = 0.2$ and $k = 5$, the adversary's advantage is dominated by $\alpha$ for any $\alpha > 2$. When using $\mathsf{LDP}$ alone (no shuffling), $\alpha = \infty$ and the adversary can exactly recover which values came from Alice's household. As such, even a moderate $\alpha$ value (obtained via $d_\sigma$-privacy) significantly reduces the ability to re-identify the values.

- When the loss is dominated by $\epsilon$ ($2k\epsilon \gg \alpha$), the above expression allows us to disentangle the *source of privacy loss*. In this regime, adversaries get most of their advantage from the bag of values released, not from the order of the release. That is, even if $\alpha = 0$ (uniform random shuffling), participants still suffer a large risk of re-identification simply due to the noisy values being reported. Thus, no shuffling mechanism can prevent re-identification in this regime.

**Discussion.** In spirit, $\mathsf{DP}$ does not guarantee protection against recovering $\mathsf{DO}_i$'s private $x_i$ value. It guarantees that – had a user not participated (or equivalently submitted a false value $x_i'$) – the adversary would have about the same ability to learn their true value, potentially from the responses of other data owners. In other words, the choice to participate is unlikely to be responsible for the disclosure of $x_i$. Similarly, $d_\sigma$-privacy does not prevent disclosure of $x_i$. By requiring indistinguishability of neighboring permutations, it guarantees that – had the data owners of any group $G_i$ completely swapped identities – the adversary would have about the same ability to learn $x_i$. So most likely, Alice's household is not uniquely responsible for a disclosure of her $x_i$: had her household swapped identities with any of her neighbors, the adversary would probably draw the same conclusion on $x_i$. Or, as detailed in Thm.4.2, an adversary cannot reliably resolve which $\{z\}$ values originated from Alice's household, so they cannot draw conclusions based on her household's responses. In a nutshell,

- Inference attacks can recover a data owner $\mathsf{DO}_i$'s private data $x_i$ from the responses of other data owners. The order of the data acts as the proxy for the data owner's identity which can aid an adversary in corralling the subset of other data owners who correlate with $\mathsf{DO}_i$ (required to make a reliable inference of $x_i$).

- DP alleviates concerns that $\mathsf{DO}_i$'s choice to share data ($y_i$) will result in disclosure of $x_i$, and $d_\sigma$-privacy alleviates concerns that $\overline{\mathsf{DO}_i}$'s group's ($G_i$) choice to share their identity will result in disclosure of $x_i$.

### 4.4 $d_\sigma$-PRIVATE SHUFFLING MECHANISM

We now describe our novel shuffling mechanism that can achieve $d_\sigma$-privacy. In a nutshell, our mechanism samples a permutation from a suitable Mallows model and shuffles the data sequence accordingly. We can characterize the $d_\sigma$-privacy guarantee of our mechanism in the same way as that of the DP guarantee of classic mechanisms (Dwork & Roth, 2014) – with variance and sensitivity. Intuitively, a larger dispersion parameter $\theta \in \mathbb{R}$ (Def. 3.2) reduces randomness over permutations, increasing utility and increasing (worsening) the privacy parameter $\alpha$. The maximum value of $\theta$ for a given $\alpha$ guarantee depends on the sensitivity of the rank distance measure $\boldsymbol{\delta}(\cdot)$ over all neighboring permutations $N_\mathcal{G}$. Formally, we define the sensitivity as

$$\Delta(\sigma_0 : \boldsymbol{\delta}, \mathcal{G}) = \max_{(\sigma,\sigma') \in N_\mathcal{G}} |\boldsymbol{\delta}(\sigma_0\sigma, \sigma_0) - \boldsymbol{\delta}(\sigma_0\sigma', \sigma_0)|,$$

the maximum change in distance $\boldsymbol{\delta}(\cdot)$ from the reference permutation $\sigma_0$ for any pair of neighboring permutations $(\sigma, \sigma') \in N_\mathcal{G}$ permuted by $\sigma_0$. The privacy parameter of the mechanism is then proportional to its sensitivity $\alpha = \theta \cdot \Delta(\sigma_0 : \boldsymbol{\delta}, \mathcal{G})$.

**Algorithm 1:** $d_\sigma$-private Shuffling Mech.

**Input:** LDP sequence $\mathbf{y} = \langle y_1, \cdots, y_n \rangle$;
 Public aux. info. $\mathbf{t} = \langle t_1, \cdots t_n \rangle$;
 Dist. threshold $r$; Priv. param. $\alpha$;

**Output:** $\mathbf{z}$ - Shuffled output sequence;

1 $\mathcal{G} = ComputeGroupAssignment\,(\mathbf{t}, r)$;

2 Construct graph $\mathbb{G}$ with
 a) vertices $V = \{1, 2, \cdots, n\}$
 b) edges $E = \{(i, j) : j \in G_i, G_i \in \mathcal{G}\}$

3 $root = \arg\max_{i \in [n]} |G_i|$;

4 $\sigma_0 = \mathsf{BFS}(\mathbb{G}, root)$;

5 $\Delta = ComputeSensitivity(\sigma_0, \mathcal{G})$

6 $\theta = \alpha/\Delta$;

7 $\hat{\sigma} \sim \mathbb{P}_{\theta, \boldsymbol{\delta}}(\sigma_0)$ ;

8 $\sigma^* = \sigma_0^{-1}\hat{\sigma}$;

9 $\mathbf{z} = \langle y_{\sigma^*(1)}, \cdots y_{\sigma^*(n)} \rangle$;

10 Return $\mathbf{z}$;

Given $\mathcal{G}$ and a reference permutation $\sigma_0$, the sensitivity of a rank distance measure $\boldsymbol{\delta}(\cdot)$ depends on the *width*, $\omega_\mathcal{G}^\sigma$, which measures how 'spread apart' the members of any group of $\mathcal{G}$ are in $\sigma_0$:

$$\omega_{G_i}^\sigma = \max_{(j,k) \in G_i \times G_i} |\sigma^{-1}(j) - \sigma^{-1}(k)|, i \in [n]; \qquad \omega_\mathcal{G}^\sigma = \max_{G_i \in \mathcal{G}} \omega_{G_i}^\sigma$$

For example, for $\sigma = (1\ 3\ 7\ 8\ 6\ 4\ 5\ 2\ 9\ 10)$ and $G_1 = \{1, 7, 8, 2, 5, 6\}$, $\omega_{G_1}^\sigma = |\sigma^{-1}(1) - \sigma^{-1}(2)| = 7$. The sensitivity is an increasing function of the width. For instance, for Kendall's $\tau$ distance $\boldsymbol{\delta}_\tau(\cdot)$ we have $\Delta(\sigma_0 : \boldsymbol{\delta}_\tau, \mathcal{G}) = \omega_\mathcal{G}^{\sigma_0}(\omega_\mathcal{G}^{\sigma_0} + 1)/2$.

If a reference permutation clusters the members of each group closely together (low width), then the groups are more likely to permute within themselves. This has two benefits. First, for the same $\theta$ ($\theta$ is an indicator of utility as it determines the dispersion of the sampled permutation), a lower value of width gives lower $\alpha$ (better privacy). Second, if a group is likely to shuffle within itself, it will have better $(\eta, \delta)$-preservation – a novel utility metric, we propose, for a shuffling mechanism. Intuitively, a mechanism is $(\eta, \delta)$-preserving w.r.t a subset of indices $S \subset [n]$ if at least $\eta\%$ of its indices are shuffled within itself with probability $(1 - \delta)$. The rationale behind this metric is that it captures the utility of the learning allowed by $d_\sigma$-privacy – if $S$ is equal to some group $G \in \mathcal{G}$, high $(\eta, \delta)$-preservation allows overall statistics of $G$ to be captured since $\eta\%$ of the correct data values remain preserved. We present the formal discussion in App. A.7.

Unfortunately, minimizing $\omega_\mathcal{G}^\sigma$ is an NP-hard problem (Thm. A.3 in App. A.9). Instead, we estimate the optimal $\sigma_0$ using the following heuristic[3] approach based on a graph breadth first search.

**Algorithm Description.** Alg. 1 above proceeds as follows. We first compute the group assignment, $\mathcal{G}$, based on the public auxiliary information and desired threshold $r$ following discussion in Sec. 4.2 (Step 1). Then we construct $\sigma_0$ with a breadth first search (BFS) graph traversal.

We translate $\mathcal{G}$ into an undirected graph $(V, E)$, where the vertices are indices $V = [n]$ and two indices $i, j$ are connected by an edge if they are both in some group (Step 2). Next, $\sigma_0$ is computed via a breadth first search traversal (Step 4) – if the $k$-th node in the traversal is $i$, then $\sigma_0(k) = i$. The rationale is that neighbors of $i$ (members of $G_i$) would be traversed in close succession. Hence, a neighboring node $j$ is likely to be traversed at some step $h$ near $k$ which means $|\sigma_0^{-1}(i) - \sigma_0^{-1}(j)| = |h - k|$ would be small (resulting in low width). Additionally, starting from the node with the highest degree (Steps 3-4) which corresponds to the largest group in $\mathcal{G}$ (lower bound for $\omega_\mathcal{G}^\sigma$ for any $\sigma$) helps to curtail the maximum width in $\sigma_0$. See App. A.16 for evaluations of this heuristic's approximation.

---

[3]The heuristics only affect $\sigma_0$ (and utility). Once $\sigma_0$ is fixed, $\Delta$ is computed exactly as discussed above.

This is followed by the computation of the dispersion parameter, $\theta$, for our Mallows model (Steps 5-6). Next, we sample a permutation from the Mallows model (Step 7) $\hat{\sigma} \sim \mathbb{P}_\theta(\sigma : \sigma_0)$ and we apply the inverse reference permutation to it, $\sigma^* = \sigma_0^{-1}\hat{\sigma}$ to obtain the desired permutation for shuffling. Recall that $\hat{\sigma}$ is (most likely) close to $\sigma_0$, which is unrelated to the original order of the data. $\sigma_0^{-1}$ therefore brings $\sigma^*$ back to a shuffled version of the original sequence (identity permutation $\sigma_I$). Note that since Alg. 1 is publicly known, the adversary/analyst knows $\sigma_0$. Hence, even in the absence of this step from our algorithm, the adversary/analyst could perform this anyway. Finally, we permute $\mathbf{y}$ according to $\sigma^*$ and output the result $\mathbf{z} = \hat{\sigma}(\mathbf{y})$ (Steps 9-10).

**Theorem 4.3.** *Alg. 1 is $(\alpha, \mathcal{G})$-$d_\sigma$ private where $\alpha = \theta \cdot \Delta(\sigma_0 : \mathfrak{d}, \mathcal{G})$.*

The proof is in App. A.11. Note that Alg. 1 provides the same level of privacy ($\alpha$) for any two group assignment $\mathcal{G}, \mathcal{G}'$ as long as they have the same sensitivity, i.e, $\Delta(\sigma_0 : \mathfrak{d}_\tau, \mathcal{G}) = \Delta(\sigma_0 : \mathfrak{d}_\tau, \mathcal{G}')$. This leads to the following theorem which generalizes the privacy guarantee for any group assignment.

**Theorem 4.4.** *Alg. 1 satisfies $(\alpha', \mathcal{G}')$-$d_\sigma$ privacy for any group assignment $\mathcal{G}'$ with $\alpha' = \alpha \frac{\Delta(\sigma_0 : \mathfrak{d}, \mathcal{G}')}{\Delta(\sigma_0 : \mathfrak{d}, \mathcal{G})}$ (proof in App. A.12.)*

**Note.** Producing $\sigma^*$ is completely data ($\mathbf{y}$) independent. It only requires access to the public auxiliary information $\mathbf{t}$. Hence, Steps $1 - 6$ can be performed in a pre-processing phase and do not contribute to the actual running time. See App. A.10 for an illustration of Alg. 1 and runtime analysis.

## 5 EVALUATION

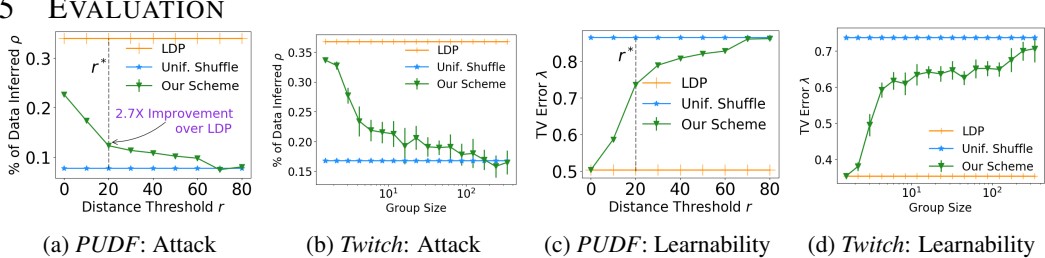

(a) *PUDF*: Attack    (b) *Twitch*: Attack    (c) *PUDF*: Learnability    (d) *Twitch*: Learnability

Figure 4: Our scheme interpolates between standard LDP (orange line) and uniform shuffling (blue line) in both privacy and data learnability. All plots increase group size along x-axis (except (d)). (a) $\rightarrow$ (b): The fraction of participants vulnerable to an inferential attack. (c) $\rightarrow$ (d): The accuracy of a calibration model trained on $\mathbf{z}$ predicting the distribution of LDP outputs at any point $t \in \mathcal{T}$, such as the distribution of medical insurance types used specifically in the Houston area (not possible when uniformly shuffling across Texas).

The previous sections describe how our shuffling framework interpolates between standard LDP and uniform random shuffling. We now experimentally evaluate this asking the following two questions –

**Q1.** Does the Alg. 1 mechanism protect against realistic inference attacks?
**Q2.** How well can Alg. 1 tune a model's ability to learn trends within the shuffled data, i.e., tune *data learnability*?

We evaluate on four datasets. We are not aware of any prior work that provides comparable local inferential privacy. Hence, we baseline our mechanism with the two extremes: standard LDP and uniform random shuffling. For concreteness, we detail our procedure with the *PUDF* dataset (PUD) (license), which comprises $n \approx 29$k psychiatric patient records from Texas. Each data owner's sensitive value $x_i$ is their medical payment method, which is reflective of socioeconomic class (such as medicaid or charity). Public auxiliary information $t \in \mathcal{T}$ is the hospital's geolocation. Such information is used for understanding how payment methods (and payment amounts) vary from town to town for insurances in practice (Eric Lopez, 2020). Uniform shuffling across Texas precludes such analyses. Standard LDP risks inference attacks, since patients attending hospitals in the same neighborhood have similar socioeconomic standing and use similar payment methods, allowing an adversary to correlate their noisy $y_i$'s. To trade these off, we apply Alg. 1 with $d(\cdot)$ being distance (km) between hospitals, $\alpha = 4$ and Kendall's $\tau$ rank distance measure for permutations.

Our inference attack predicts $\mathsf{DO}_i$'s $x_i$ by taking a majority vote of the $z_j$ values of the 25 data owners within $r^*$ of $t_i$ and who are most similar to $\mathsf{DO}_i$ w.r.t some additional privileged auxiliary information $t_j^p \in \mathcal{T}_p$. For PUDF, this includes the 25 data owners who attended hospitals that are within $r^*$ km of $\mathsf{DO}_i$'s hospital, and are most similar in payment amount $t_j^p$. Using an $\epsilon = 2.5$ randomized response mechanism, we resample the LDP sequence $\mathbf{y}$ 50 times, and apply Alg. 1's chosen permutation to

each, producing 50 $\mathbf{z}$'s. We then mount the majority vote attack on each $x_i$ for each $\mathbf{z}$. If the attack on a given $x_i$ is successful across $\geq 90\%$ of these LDP trials, we mark that data owner as vulnerable – although they randomize with LDP, there is a $\geq 90\%$ chance that a simple inference attack can recover their true value. We record the fraction of vulnerable data owners as $\rho$. We report 1-standard deviation error bars over 10 trials.

Additionally, we evaluate *data learnability* – how well the underlying statistics of the dataset are preserved across $\mathcal{T}$. For *PUDF*, this means training a model on the shuffled $\mathbf{z}$ to predict the distribution of payment methods used near, for instance, $t_i =$ Houston for $\mathsf{DO}_i$. For this, we train a calibrated model, $\mathtt{Cal} : \mathcal{T} \to \mathcal{D}_x$, on the shuffled outputs where $\mathcal{D}_x$ is the set of all distributions on the domain of sensitive attributes $\mathcal{X}$. We implement $\mathtt{Cal}$ as a gradient boosted decision tree (GBDT) model (Friedman, 2001) calibrated with Platt scaling (Niculescu-Mizil & Caruana, 2005). For each location $t_i$, we treat the empirical distribution of $x_i$ values within $r^*$ as the ground truth distribution at $t_i$, denoted by $\mathcal{E}(t_i) \in \mathcal{D}_x$. Then, for each $t_i$, we measure the Total Variation error between the predicted and ground truth distributions $\mathrm{TV}\big(\mathcal{E}(t_i), \mathtt{Cal}_r(t_i)\big)$. We then report $\lambda(r)$ – the average TV error for distributions predicted at each $t_i \in \mathbf{t}$ normalized by the TV error of naively guessing the uniform distribution at each $t_i$. With standard LDP, this task can be performed relatively well at the risk of inference attacks. With uniformly shuffled data, it is impossible to make geographically localized predictions unless the distribution of payment methods is identical in every Texas locale.

We additionally perform the above experiments on the following three datasets

- *Twitch* (Rozemberczki et al., 2019). This dataset, gathered from the *Twitch* social media platform, includes a graph of $\approx 9K$ edges (mutual friendships) along with node features. The user's history of explicit language is private $\mathcal{X} = \{0, 1\}$. $\mathcal{T}$ is a user's mutual friendships, i.e. $t_i$ is the $i$'th row of the graph's adjacency matrix. We do not have any $\mathcal{T}_P$ here and select the 25 neighbors randomly.

- *Syn*. This is a synthetic dataset of size $20K$ which can be classified at three granularities – 8-way, 4-way and 2-way (Fig. 1a shows a scaled down version of the dataset). The eight color labels are private $\mathcal{X} = [8]$; the 2D-positions are public $\mathcal{T} = \mathbb{R}^2$. For learnability, we measure the accuracy of 8-way, 4-way and 2-way GBDT models trained on $\mathbf{z}$ on an equal sized test set at each $r$.

- *Adult* (Dua & Graff, 2017). This dataset is derived from the 1994 Census and has $\approx 33K$ records. Whether $\mathsf{DO}_i$'s annual income is $\geq 50k$ is considered private, $\mathcal{X} = \{\geq 50k, < 50k\}$. $\mathcal{T} = [17, 90]$ is age and $\mathcal{T}_P$ is the individual's marriage status. Due to lack of space figures are in App. A.14.2.

**Experimental Results.**

**Q1.** Our formal guarantee on the inferential privacy loss (Thm. 4.1) is described w.r.t to a 'strong' adversary (with access to $\{y_{G_i}\}, \mathbf{y}_{\overline{G}_i}$). Here, we test how well does our proposed scheme (Alg. 1) protect against inference attacks on real-world datasets without any such assumptions. Additionally, to make our attack more realistic, the adversary has access to extra privileged auxiliary information $\mathcal{T}_P$ which is *not used* by Alg. 10. Fig. 4a$\to$ 4b show that our scheme significantly reduces the attack efficacy. For instance, $\rho$ is reduced by $2.7X$ at the attack distance threshold $r^*$ for *PUDF*.

Additionally, $\rho$ for our scheme varies from that of LDP[4] (minimum privacy) to uniform shuffle (maximum privacy) with increasing $r$ (equivalently group size as in Fig. 4b) thereby spanning the entire privacy spectrum. As expected, $\rho$ decreases with decreasing privacy parameter $\alpha$ (Fig. 8b).

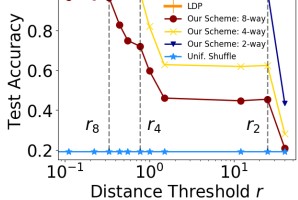

Figure 5: *Syn*: Learnability

**Q2.** Fig.4c $\to$ 4d show that $\lambda$ varies from that of LDP (maximum learnability) to that of uniform shuffle (minimum learnability) with increasing $r$ (equivalently, group size), thereby providing tunability. Interestingly, for *Adult* our scheme reduces $\rho$ by $1.7X$ at the same $\lambda$ as that of LDP for $r = 1$ (Fig. 8c). Fig. 5 shows that the distance threshold $r$ defines the granularity at which the data can be classified. LDP allows 8-way classification while uniform shuffling allows none. The granularity of classification can be tuned by our scheme – $r_8$, $r_4$ and $r_2$ mark the thresholds for 8-way, 4-way and 2-way classifications, respectively.

## 6 CONCLUSION

We have proposed a new privacy definition, $d_\sigma$-privacy that casts new light on the inferential privacy benefits of shuffling and a novel shuffling mechanism to achieve the same.

---

[4]Our scheme gives lower $\rho$ than LDP at $r = 0$ because the resulting groups are non-singletons. For instance, for PUDF, $G_i$ includes all individuals with the same zipcode as $\mathsf{DO}_i$.

## 7 ETHIS STATEMENT

It is the aim of this paper to formalize the enhanced privacy guarantees offered by shuffling, to provide intuition of what those formal guarantees semantically offer to data owners, and to demonstrate an algorithm + experiments which offer these guarantees while meeting analyst utility requirements. We feel that all of these aims as well as the public datasets used are ethical.

## 8 REPRODUCIBILITY STATEMENT

The majority of this paper formalizes a novel perspective on the privacy guarantees achieved by shuffling (i.e. randomizing the order of the data as opposed to the values). Detailed proofs as well as intuitive discussions are provided in the Appendix. All datasets are public. A `.zip` file demonstrating code of each experiment has been uploaded as supplementary material.

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

# A APPENDIX

## A.1 BACKGROUND CNTD.

## A.2 LOCAL INFERENTIAL PRIVACY

Local inferential privacy captures what information a Bayesian adversary Kifer & Machanavajjhala (2014), with some prior, can learn in the LDP setting. Specifically, it measures the largest possible ratio between the adversary's posterior and prior beliefs about an individual's data after observing a mechanism's output .

**Definition A.1.** (Local Inferential Privacy Loss Kifer & Machanavajjhala (2014)) Let $\mathbf{x} = \langle x_1, \cdots, x_n \rangle$ and let $\mathbf{y} = \langle y_1, \cdots, y_n \rangle$ denote the input (private) and output sequences (observable to the adversary) in the LDP setting. Additionally, the adversary's auxiliary knowledge is modeled by a prior distribution $\mathcal{P}$ on $\mathbf{x}$. The inferential privacy loss for the input sequence $\mathbf{x}$ is given by

$$\mathbb{L}_{\mathcal{P}}(\mathbf{x}) = \max_{\substack{i \in [n] \\ a,b \in \mathcal{X}}} \left( \log \frac{\Pr_{\mathcal{P}}[\mathbf{y}|x_i = a]}{\Pr_{\mathcal{P}}[\mathbf{y}|x_i = b]} \right) = \max_{\substack{i \in [n] \\ a,b \in \mathcal{X}}} \left( \left| \log \frac{\Pr_{\mathcal{P}}[x_i = a|\mathbf{y}]}{\Pr_{\mathcal{P}}[x_i = b|\mathbf{y}]} - \log \frac{\Pr_{\mathcal{P}}[x_i = a]}{\Pr_{\mathcal{P}}[x_i = b]} \right| \right) \quad (4)$$

Bounding $\mathbb{L}_{\mathcal{P}}(\mathbf{x})$ would imply that the adversary's belief about the value of any $x_i$ does not change by much even after observing the output sequence $\mathbf{y}$. This means that an informed adversary does not learn much about the individual $i$'s private input upon observation of the entire private dataset $\mathbf{y}$.

Here we define two rank distance measures

**Definition A.2** (Kendall's $\tau$ Distance). For any two permutations, $\sigma, \pi \in \mathrm{S}_n$, the Kendall's $\tau$ distance $\mathfrak{d}_\tau(\sigma, \pi)$ counts the number of pairwise disagreements between $\sigma$ and $\pi$, i.e., the number of item pairs that have a relative order in one permutation and a different order in the other. Formally,

$$\mathfrak{d}_\tau(\sigma, \pi) = \Big| \big\{ (i,j) : i < j, \big[\sigma(i) > \sigma(j) \wedge \pi(i) < \pi(j)\big]$$

$$\vee \big[\sigma(i) < \sigma(j) \wedge \pi(i) > \pi(j)\big] \big\} \Big| \quad (5)$$

For example, if $\sigma = (1\ 2\ 3\ 4\ 5\ 6\ 7\ 8\ 9\ 10)$ and $\pi = (1\ 2\ 3\ \underline{6}\ 5\ \underline{4}\ 7\ 8\ 9\ 10)$, then $\mathfrak{d}_\tau(\sigma, \pi) = 3$.

Next, Hamming distance measure is defined as follows.

**Definition A.3** (Hamming Distance). For any two permutations, $\sigma, \pi \in \mathrm{S}_n$, the Hamming distance $\mathfrak{d}_H(\sigma, \pi)$ counts the number of positions in which the two permutations disagree. Formally,

$$\mathfrak{d}_H(\sigma, \pi) = \Big| \big\{ i \in [n] : \sigma(i) \neq \pi(i) \big\} \Big|$$

Repeating the above example, if $\sigma = (1\ 2\ 3\ 4\ 5\ 6\ 7\ 8\ 9\ 10)$ and $\pi = (1\ 2\ 3\ \underline{6}\ 5\ \underline{4}\ 7\ 8\ 9\ 10)$, then $\mathfrak{d}_H(\sigma, \pi) = 2$.

## A.3 $d_\sigma$-PRIVACY AND THE DE FINETTI ATTACK

We now show that a strict instance of $d_\sigma$privacy is sufficient for thwarting any de Finetti attack Kifer (2009) on individuals. The de Finetti attack involves a Bayesian adversary, who, assuming some degree of correlation between data owners, attempts to recover the true permutation from the shuffled data. As written, the de Finetti attack assumes the sequence of sensitive attributes and side information $(x_1, t_1), \ldots, (x_n, t_n)$ are *exchangeable*: any ordering of them is equally likely. By the de Finetti theorem, this implies that they are i.i.d. conditioned on some latent measure $\theta$. To balance privacy with utility, the $\mathbf{x}$ sequence is non-uniformly randomly shuffled w.r.t. the $\mathbf{t}$ sequence producing a shuffled sequence $\mathbf{z}$, which the adversary observes. Conditioning on $\mathbf{z}$ the adversary updates their posterior on $\theta$ (i.e. posterior on a model predicting $x_i|t_i$), and thereby their posterior

predictive on the true $\mathbf{x}$. The definition of privacy in Kifer (2009) holds that the adversary's posterior beliefs are close to their prior beliefs by some metric on distributions in $\mathcal{X}$, $\delta(\cdot, \cdot)$:

$$\delta\Big(\Pr[x_i], \Pr[x_i|\mathbf{z}]\Big) \leq \alpha$$

We now translate the de Finetti attack to our setting. First, to align notation with the rest of the paper we provide privacy to the sequence of LDP values $\mathbf{y}$ since we shuffle those instead of the $\mathbf{x}$ values as in Kifer (2009). We use max divergence (multiplicative bound on events used in DP ) for $\delta$:

$$\Pr[y_i \in O] \leq e^\alpha \Pr[y_i \in O|\mathbf{z}]$$

$$\Pr[y_i \in O|\mathbf{z}] \leq e^\alpha \Pr[y_i \in O]$$

which, for compactness, we write as

$$\Pr[y_i \in O] \approx_\alpha \Pr[y_i \in O|\mathbf{z}] \quad . \tag{6}$$

We restrict ourselves to shuffling mechanisms, where we only randomize the order of sensitive values. By learning the unordered values $\{y\}$ alone, an adversary may have arbitrarily large updates to its posterior (e.g. if all values are identical), breaking the privacy requirement above. With this in mind, we assume the adversary already knows the unordered sequence of values $\{y\}$ (which they will learn anyway), and has a prior on permutations $\sigma$ allocating values from that sequence to individuals. We then generalize the de Finetti problem to an adversary with an *arbitrary* prior on the true permutation $\sigma$, and observes a randomize permutation $\sigma'$ from the shuffling mechanism. We require that the adversary's prior belief that $\sigma(i) = j$ is close to their posterior belief for all $i, j \in [n]$:

$$\Pr[\sigma \in \Sigma_{i,j}] \approx_\alpha \Pr[\sigma \in \Sigma_{i,j}|\sigma'] \quad \forall i,j \in [n], \forall \sigma' \in S_n \quad , \tag{7}$$

where $\Sigma_{i,j} = \{\sigma \in S_n : \sigma(i) = j\}$, the set of permutations assigning element $j$ to $\mathsf{DO}_i$. Conditioning on any unordered sequence $\{y\}$ with all unique values, the above condition is necessary to satisfy Eq. equation 6 for events of the form $O = \{y_i = a\}$, since $\{y_i = a\} = \{\Sigma_{i,j}\}$ for some $j \in [n]$. For any $\{y\}$ with repeat values, it is sufficient since $\Pr[y_i = a]$ is the sum of probabilities of disjoint events of the form $\Pr[\sigma \in \Sigma_{i,k}]$ for various $k \in [n]$ values.

We now show that a strict instance of $d_\sigma$-privacy satisfies Eq. equation 7. Let $\widehat{\mathcal{G}}$ be any group assignment such that at least one $G_i \in \widehat{\mathcal{G}}$ includes all data owners, $G_i = \{1, 2, \ldots, n\}$.

**Property 1.** *A $(\widehat{\mathcal{G}}, \alpha)$-$d_\sigma$-private shuffling mechanism $\sigma' \sim \mathcal{A}$ satisfies*

$$\Pr[\sigma \in \Sigma_{i,j}] \approx_\alpha \Pr[\sigma \in \Sigma_{i,j}|\sigma']$$

*for all $i, j \in [n]$ and all priors on permutations $\Pr[\sigma]$.*

*Proof.*

**Lemma 1.** *For any prior $\Pr[\sigma]$, Eq. equation 7 is equivalent to the condition*

$$\frac{\sum_{\hat{\sigma} \in \overline{\Sigma}_{i,j}} \Pr[\hat{\sigma}] \Pr[\sigma'|\hat{\sigma}]}{\sum_{\hat{\sigma} \in \Sigma_{i,j}} \Pr[\hat{\sigma}] \Pr[\sigma'|\hat{\sigma}]} \approx_\alpha \frac{\sum_{\hat{\sigma} \in \overline{\Sigma}_{i,j}} \Pr[\hat{\sigma}]}{\sum_{\hat{\sigma} \in \Sigma_{i,j}} \Pr[\hat{\sigma}]} \tag{8}$$

*where the set $\overline{\Sigma}_{i,j}$ is the complement of $\Sigma_{i,j}$.*

Under grouping $\hat{\mathcal{G}}$, every permutation $\sigma_a \in \Sigma_{i,j}$ neighbors every permutation $\sigma_b \in \overline{\Sigma}_{i,j}$, $\sigma_a \approx_{\hat{\mathcal{G}}} \sigma_b$, for any $i, j$. By the definition of $d_\sigma$-privacy, we have that for any observed permutation $\sigma'$ output by the mechanism:

$$\Pr[\sigma'|\sigma = \sigma_a] \approx_\alpha \Pr[\sigma'|\sigma = \sigma_b] \quad \forall \sigma_a \in \Sigma_{i,j}, \sigma_b \in \overline{\Sigma}_{i,j}, \sigma' \in S_n \quad .$$

This implies Eq. 8. Thus, $(\widehat{\mathcal{G}}, \alpha)$-$d_\sigma$-privacy implies Eq. 8, which implies Eq. 7, thus proving the property. □

Using Lemma 1, we may also show that this strict instance of $d_\sigma$-privacy is *necessary* to block all de Finetti attacks:

**Property 2.** *A $(\widehat{\mathcal{G}}, \alpha)$-$d_\sigma$-private shuffling mechanism $\sigma' \sim \mathcal{A}$ is necessary to satisfy*

$$\Pr[\sigma \in \Sigma_{i,j}] \approx_\alpha \Pr[\sigma \in \Sigma_{i,j}|\sigma']$$

*for all $i, j \in [n]$ and all priors on permutations $\Pr[\sigma]$.*

*Proof.* If our mechanism $\mathcal{A}$ is not $(\widehat{\mathcal{G}}, \alpha)$-$d_\sigma$-private, then for some pair of true (input) permutations $\sigma_a \neq \sigma_b$ and some released permutation $\sigma' \sim \mathcal{A}$, we have that

$$\Pr[\sigma'|\sigma_b] \geq e^\alpha \Pr[\sigma'|\sigma_a] \quad .$$

Under $\hat{\mathcal{G}}$, all permutations neighbor each other, so $\sigma_a \approx_{\hat{\mathcal{G}}} \sigma_b$. Since $\sigma_a \neq \sigma_b$, then for some $i, j \in [n]$, $\sigma_a \in \Sigma_{i,j}$ and $\sigma_b \in \overline{\Sigma}_{i,j}$: one of the two permutations assigns some $j$ to some $\mathsf{DO}_i$ and the other does not. Given this, we may construct a bimodal prior on the true $\sigma$ that assigns half its probability mass to $\sigma_a$ and the rest to $\sigma_b$,

$$\Pr[\sigma_a] = \Pr[\sigma_b] = \frac{1}{2} \quad .$$

Therefore, for released permutation $\sigma'$, the RHS of Eq. 8 is 1, and the LHS is

$$\frac{\sum_{\hat{\sigma} \in \overline{\Sigma}_{i,j}} \Pr[\hat{\sigma}] \Pr[\sigma'|\hat{\sigma}]}{\sum_{\hat{\sigma} \in \Sigma_{i,j}} \Pr[\hat{\sigma}] \Pr[\sigma'|\hat{\sigma}]} = \frac{1/2 \Pr[\sigma'|\sigma_b]}{1/2 \Pr[\sigma'|\sigma_a]}$$
$$\geq e^{\alpha} \quad ,$$

violating Eq. 8, thus violating Eq. 7, and failing to prevent de Finetti attacks against this bimodal prior. $\qquad\square$

Ultimately, unless we satisfy $d_\sigma$-privacy shuffling the entire dataset, there exists some prior on the true permutation $\Pr[\sigma]$ such that after observing the shuffled $\mathbf{z}$ permuted by $\sigma'$, the adversary's posterior belief on one permutation is larger than their prior belief by a factor $\geq e^{\alpha}$. If we suppose that the set of values $\{y\}$ are all distinct, this means that for some $a \in \{y\}$, the adversary's belief that $y_i = a$ is significantly larger after observing $\mathbf{z}$ than it was before.

Now to prove Lemma 1:

*Proof.*

$$\Pr[\sigma \in \Sigma_{i,j}] \approx_\alpha \Pr[\sigma \in \Sigma_{i,j}|\sigma']$$
$$\Pr[\sigma \in \Sigma_{i,j}] \approx_\alpha \frac{\Pr[\sigma'|\sigma \in \Sigma_{i,j}] \Pr[\sigma \in \Sigma_{i,j}]}{\sum_{\hat{\sigma} \in S_n} \Pr[\hat{\sigma}] \Pr[\sigma'|\hat{\sigma}]}$$
$$\sum_{\hat{\sigma} \in S_n} \Pr[\hat{\sigma}] \Pr[\sigma'|\hat{\sigma}] \approx_\alpha \Pr[\sigma'|\sigma \in \Sigma_{i,j}]$$
$$\sum_{\hat{\sigma} \in S_n} \Pr[\hat{\sigma}] \Pr[\sigma'|\hat{\sigma}] \approx_\alpha \Pr[\sigma \in \Sigma_{i,j}]^{-1} \sum_{\hat{\sigma} \in \Sigma_{i,j}} \Pr[\hat{\sigma}] \Pr[\sigma'|\hat{\sigma}]$$
$$\sum_{\hat{\sigma} \in \Sigma_{i,j}} \Pr[\hat{\sigma}] \Pr[\sigma'|\hat{\sigma}] + \sum_{\hat{\sigma} \in \overline{\Sigma}_{i,j}} \Pr[\hat{\sigma}] \Pr[\sigma'|\hat{\sigma}] \approx_\alpha \Pr[\sigma \in \Sigma_{i,j}]^{-1} \sum_{\hat{\sigma} \in \Sigma_{i,j}} \Pr[\hat{\sigma}] \Pr[\sigma'|\hat{\sigma}]$$
$$\sum_{\hat{\sigma} \in \overline{\Sigma}_{i,j}} \Pr[\hat{\sigma}] \Pr[\sigma'|\hat{\sigma}] \approx_\alpha \sum_{\hat{\sigma} \in \Sigma_{i,j}} \Pr[\hat{\sigma}] \Pr[\sigma'|\hat{\sigma}]\Big(\frac{1}{\Pr[\sigma \in \Sigma_{i,j}]} - 1\Big)$$
$$\frac{\sum_{\hat{\sigma} \in \overline{\Sigma}_{i,j}} \Pr[\hat{\sigma}] \Pr[\sigma'|\hat{\sigma}]}{\sum_{\hat{\sigma} \in \Sigma_{i,j}} \Pr[\hat{\sigma}] \Pr[\sigma'|\hat{\sigma}]} \approx_\alpha \frac{\sum_{\hat{\sigma} \in \overline{\Sigma}_{i,j}} \Pr[\hat{\sigma}]}{\sum_{\hat{\sigma} \in \Sigma_{i,j}} \Pr[\hat{\sigma}]}$$

$\qquad\square$

As such, a strict instance of $d_\sigma$-privacy can defend against any de Finetti attack (i.e. for any prior $\Pr[\sigma]$ on permutations), wherein at least one group $G_i \in \mathcal{G}$ includes all data owners. Furthermore, it is necessary. This makes sense. In order to defend against any prior, we need to significantly shuffle the entire dataset. Without a restriction of priors as in Pufferfish Kifer & Machanavajjhala (2014), the de Finetti attack (i.e. uninformed Bayesian adversaries) is an indelicate metric for evaluating the privacy of shuffling mechanisms: to achieve significant privacy, we must sacrifice all utility. This in many regards is reminiscent of the no free lunch for privacy theorem established in Kifer & Machanavajjhala (2011). As such, there is a need for more flexible privacy definitions for shuffling mechanisms.

## A.4 ADDITIONAL PROPERTIES OF $d_\sigma$-PRIVACY

**Lemma 2** (Convexity). *Let $\mathcal{A}_1, \ldots \mathcal{A}_k : \mathcal{Y}^n \mapsto \mathcal{V}$ be a collection of $k$ $(\alpha, \mathcal{G})$-$d_\sigma$private mechanisms for a given group assignment $\mathcal{G}$ on a set of $n$ entities. Let $\mathcal{A} : \mathcal{Y}^n \mapsto \mathcal{V}$ be a convex combination of these $k$ mechanisms, where the probability of releasing the output of mechanism $\mathcal{A}_i$ is $p_i$, and $\sum_{i=1}^k p_i = 1$. $\mathcal{A}$ is also $(\alpha, \mathcal{G})$-$d_\sigma$private w.r.t. $\mathcal{G}$.*

*Proof.* For any $(\sigma, \sigma') \in N_{\mathcal{G}}$ and $\mathbf{y} \in \mathcal{Y}$:

$$\Pr[\mathcal{A}(\sigma(\mathbf{y})) \in O] = \sum_{i=1}^{k} p_i \Pr[\mathcal{A}_i(\sigma(\mathbf{y})) \in O]$$

$$\leq e^{\alpha} \sum_{i=1}^{k} p_i \Pr[\mathcal{A}_i(\sigma'(\mathbf{y})) \in O]$$

$$= \Pr[\mathcal{A}(\sigma'(\mathbf{y})) \in O]$$

$\square$

**Theorem A.1** (Post-processing). *Let $\mathcal{A} : \mathcal{Y}^n \mapsto \mathcal{V}$ be $(\alpha, \mathcal{G})$-$d_\sigma$private for a given group assignment $\mathcal{G}$ on a set of $n$ entities. Let $f : \mathcal{V} \mapsto \mathcal{V}'$ be an arbitrary randomized mapping. Then $f \circ \mathcal{A} : \mathcal{Y}^n \mapsto \mathcal{V}'$ is also $(\alpha, \mathcal{G})$-$d_\sigma$private.*

*Proof.* Let $g : \mathcal{V} \to \mathcal{V}'$ be a deterministic, measurable function. For any output event $\mathcal{Z} \subset \mathcal{V}'$, let $\mathcal{W}$ be its preimage:
$\mathcal{W} = \{v \in \mathcal{V} | g(v) \in \mathcal{Z}\}$. Then, for any $(\sigma, \sigma') \in N_{\mathcal{G}}$,

$$\Pr\left[g\left(\mathcal{A}(\sigma(\mathbf{y}))\right) \in \mathcal{Z}\right] = \Pr\left[\mathcal{A}(\sigma(\mathbf{y})) \in \mathcal{W}\right]$$

$$\leq e^{\alpha} \cdot \Pr\left[\mathcal{A}(\sigma'(\mathbf{y})) \in \mathcal{W}\right]$$

$$= e^{\alpha} \cdot \Pr\left[g\left(\mathcal{A}(\sigma'(\mathbf{y}))\right) \in \mathcal{Z}\right]$$

This concludes our proof because any randomized mapping can be decomposed into a convex combination of measurable, deterministic functions Dwork & Roth (2014), and as Lemma 2 shows, a convex combination of $(\alpha, \mathcal{G})$-$d_\sigma$private mechanisms is also $(\alpha, \mathcal{G})$-$d_\sigma$private. $\square$

**Theorem A.2** (Sequential Composition). *If $\mathcal{A}_1$ and $\mathcal{A}_2$ are $(\alpha_1, \mathcal{G})$- and $(\alpha_2, \mathcal{G})$-$d_\sigma$private mechanisms, respectively, that use independent randomness, then releasing the outputs $\left(\mathcal{A}_1(\mathbf{y}), \mathcal{A}_2(\mathbf{y})\right)$ satisfies $(\alpha_1 + \alpha_2, \mathcal{G})$-$d_\sigma$privacy.*

*Proof.* We have that $\mathcal{A}_1 : \mathcal{Y}^n \to \mathcal{V}'$ and $\mathcal{A}_1 : \mathcal{Y}^n \to \mathcal{V}''$ each satisfy $d_\sigma$-privacy for different $\alpha$ values. Let $\mathcal{A} : \mathcal{Y}^n \to (\mathcal{V}' \times \mathcal{V}'')$ output $\left(\mathcal{A}_1(\mathbf{y}), \mathcal{A}_2(\mathbf{y})\right)$. Then, we may write any event $\mathcal{Z} \in (\mathcal{V}' \times \mathcal{V}'')$ as $\mathcal{Z}' \times \mathcal{Z}''$, where $\mathcal{Z}' \in \mathcal{V}'$ and $\mathcal{Z}'' \in \mathcal{V}''$. We have for any $(\sigma, \sigma') \in N_{\mathcal{G}}$,

$$\Pr\left[\mathcal{A}(\sigma(\mathbf{y})) \in \mathcal{Z}\right] = \Pr\left[\left(\mathcal{A}_1(\sigma(\mathbf{y})), \mathcal{A}_2(\sigma(\mathbf{y}))\right) \in \mathcal{Z}\right]$$

$$= \Pr\left[\{\mathcal{A}_1(\sigma(\mathbf{y})) \in \mathcal{Z}'\} \cap \{\mathcal{A}_2(\sigma(\mathbf{y})) \in \mathcal{Z}''\}\right]$$

$$= \Pr\left[\{\mathcal{A}_1(\sigma(\mathbf{y})) \in \mathcal{Z}'\}\right] \Pr\left[\{\mathcal{A}_2(\sigma(\mathbf{y})) \in \mathcal{Z}''\}\right]$$

$$\leq e^{\alpha_1+\alpha_2} \Pr\left[\{\mathcal{A}_1(\sigma'(\mathbf{y})) \in \mathcal{Z}'\}\right] \Pr\left[\{\mathcal{A}_2(\sigma'(\mathbf{y})) \in \mathcal{Z}''\}\right]$$

$$= e^{\alpha_1+\alpha_2} \cdot \Pr\left[\mathcal{A}(\sigma'(\mathbf{y})) \in \mathcal{Z}\right]$$

$\square$

## A.5 PROOF FOR THM. 4.1

**Theorem 4.1** *For a given group assignment $\mathcal{G}$ on a set of $n$ data owners, if a shuffling mechanism $\mathcal{A} : \mathcal{Y}^n \mapsto \mathcal{Y}^n$ is $(\alpha, \mathcal{G})$-$d_\sigma$private, then for each data owner $DO_i, i \in [n]$,*

$$\max_{\substack{i \in [n] \\ a,b \in \mathcal{X}}} \left| \log \frac{\Pr_{\mathcal{P}}[x_i = a | \mathbf{z}, \{y_{G_i}\}, \mathbf{y}_{\overline{G}_i}]}{\Pr_{\mathcal{P}}[x_i = b | \mathbf{z}, \{y_{G_i}\}, \mathbf{y}_{\overline{G}_i}]} - \log \frac{\Pr_{\mathcal{P}}[x_i = a | \{y_{G_i}\}, \mathbf{y}_{\overline{G}_i}]}{\Pr_{\mathcal{P}}[x_i = b | \{y_{G_i}\}, \mathbf{y}_{\overline{G}_i}]} \right| \leq \alpha$$

*for a prior distribution $\mathcal{P}$, where $\mathbf{z} = \mathcal{A}(\mathbf{y})$ and $\mathbf{y}_{\overline{G}_i}$ is the noisy sequence for data owners outside $G_i$.*

*Proof.* We prove the above by bounding the following equivalent expression for any $i \in [n]$ and $a, b \in \mathcal{X}$.

$$\frac{\mathrm{Pr}_{\mathcal{P}}[\mathbf{z}|x_i = a, \{y_{G_i}\}, \mathbf{y}_{\overline{G}_i}]}{\mathrm{Pr}_{\mathcal{P}}[\mathbf{z}|x_i = b, \{y_{G_i}\}, \mathbf{y}_{\overline{G}_i}]}$$

$$= \frac{\int \mathrm{Pr}_{\mathcal{P}}[\mathbf{y}|x_i = a, \{y_{G_i}\}, \mathbf{y}_{\overline{G}_i}] \mathrm{Pr}_{\mathcal{A}}[\mathbf{z}|\mathbf{y}] d\mathbf{y}}{\int \mathrm{Pr}_{\mathcal{P}}[\mathbf{y}|x_i = b, \{y_{G_i}\}, \mathbf{y}_{\overline{G}_i}] \mathrm{Pr}_{\mathcal{A}}[\mathbf{z}|\mathbf{y}] d\mathbf{y}}$$

$$= \frac{\sum_{\sigma \in \mathrm{S}_r} \mathrm{Pr}_{\mathcal{P}}[\sigma(\mathbf{y}^*_{G_i})|x_i = a, \mathbf{y}_{\overline{G}_i}] \mathrm{Pr}_{\mathcal{A}}[\mathbf{z}|\sigma(\mathbf{y}^*_{G_i}), \mathbf{y}_{\overline{G}_i}]}{\sum_{\sigma \in \mathrm{S}_r} \mathrm{Pr}_{\mathcal{P}}[\sigma(\mathbf{y}^*_{G_i})|x_i = b, \mathbf{y}_{\overline{G}_i}] \mathrm{Pr}_{\mathcal{A}}[\mathbf{z}|\sigma(\mathbf{y}^*_{G_i}), \mathbf{y}_{\overline{G}_i}]}$$

$$\leq \max_{\{\sigma, \sigma' \in \mathrm{S}_r\}} \frac{\mathrm{Pr}_{\mathcal{A}}[\mathbf{z}|\sigma(\mathbf{y}^*_{G_i}), \mathbf{y}_{\overline{G}_i}]}{\mathrm{Pr}_{\mathcal{A}}[\mathbf{z}|\sigma'(\mathbf{y}^*_{G_i}), \mathbf{y}_{\overline{G}_i}]}$$

$$\leq \max_{\{\sigma, \sigma' \in \mathrm{N}_{G_i}\}} \frac{\mathrm{Pr}_{\mathcal{A}}[\mathbf{z}|\sigma(\mathbf{y})]}{\mathrm{Pr}_{\mathcal{A}}[\mathbf{z}|\sigma'(\mathbf{y})]}$$

$$\leq e^{\alpha}$$

The second line simply marginalizes out the full noisy sequence $\mathbf{y}$. The third line reduces this to a sum over permutations of of $\mathbf{y}_{G_i}$, where $r = |G_i|$ and $\mathbf{y}^*_{G_i}$ is any fixed permutation of values $\{y_{G_i}\}$. This is possible since we are given the values outside the group, $\mathbf{y}_{\overline{G}_i}$, and the unordered set of values inside the group, $\{y_{G_i}\}$. Note that the permutations $\sigma$ written here are possible permutations of the LDP input, not permutations output by the mechanism applied to the input as sometimes written in other parts of this document.

The fourth line uses the fact that the numerator and denominator are both convex combinations of $\mathrm{Pr}_{\mathcal{A}}[\mathbf{z}|\sigma(\mathbf{y}^*_{G_i}), \mathbf{y}_{\overline{G}_i}]$ over all $\sigma \in \mathrm{S}_r$.

The fifth line uses the fact that for any $\mathbf{y}_{\overline{G}_i}$,

$$(\sigma(\mathbf{y}^*_{G_i}), \mathbf{y}_{\overline{G}_i}) \approx_{G_i} (\sigma'(\mathbf{y}^*_{G_i}), \mathbf{y}_{\overline{G}_i}) .$$

This allows a further upper bound over all neighboring sequences w.r.t. $G_i$, and thus over any permutation of $\mathbf{y}_{\overline{G}_i}$, as long as it is the same in the numerator and denominator. $\square$

**Discussion** The above Bayesian analysis measures what can be learned about $\mathrm{DO}_i$'s $x_i$ from observing the private release $\mathbf{z}$ relative to some other known information (the conditioned information). Under $d_\sigma$-privacy, we condition on the bag of LDP values in Alice's group $\{y_{G_i}\}$ as well as the sequence (order and value) of LDP values outside her group $\mathbf{y}_{\overline{G}_i}$. This implies that releasing the shuffled sequence $\mathbf{z}$ cannot provide much more information about Alice's $x_i$ than would releasing the LDP values outside her neighborhood (her group) and the unordered bag of LDP values inside her neighborhood, regardless of the adversary's prior knowledge $\mathcal{P}$. This is a communicable guarantee: if Alice feels comfortable with the data collection knowing that her entire neighborhood's responses will be uniformly shuffled together (including those of her household), then she ought to be comfortable with $d_\sigma$-privacy. Now, we have to provide this guarantee to Bob, a neighbor of Alice, as well as Luis, a neighbor of Bob but *not* of Alice. Thus, Bob, Alice and Luis have *distinct* and *overlapping* groups (neighborhoods). Hence, the trivial solution of uniformly shuffling the noisy responses of every group separately does not work in this case. $d_\sigma$-privacy, however, offers the above guarantee to each user (knowing that their entire neighborhood is *nearly* uniformly shuffled) while still maintaining utility (estimate disease prevalence within neighborhoods). Semantically, this is very powerful, since it implies that the noisy responses specific to one's household cannot be leveraged to infer one's disease state $x_i$.

## A.6 PROOF OF THEOREM 4.2

**Theorem 4.2**

*For $\mathcal{A}(\mathcal{M}(\mathbf{x})) = \mathbf{z}$ where $\mathcal{M}(\cdot)$ is $\epsilon$-LDP and $\mathcal{A}(\cdot)$ is $\alpha$ - $d_\sigma$ private, we have*

$$\mathrm{Pr}[\mathcal{D}_{Adv} \text{ loses}] \geq \lfloor \frac{r-k}{k} \rfloor e^{-(2k\epsilon + \alpha)} \cdot \mathrm{Pr}[\mathcal{D}_{Adv} \text{ wins}]$$

*for any input subgroup $I \subset G_i, r = |G_i|$ and $k < r/2$.*

*Proof.* We first focus on deterministic adversaries and then expand to randomized adversaries afterwards using the fact that randomized adversaries are mixtures of deterministic ones.

Our adversary $\mathcal{D}_{Adv}$ is then defined by a deterministic decision function $\eta : \mathcal{Y}^n \to [n]^k$. Upon observing $\mathbf{z}$, $\eta(\mathbf{z})$ selects $k$ elements in $\mathbf{z}$ which it believes originated from $I \subset G_i$.

In the following, let $\mathrm{Pr}_{\mathbf{z}}$ be the probability of events conditioned on the shuffled output sequence $\mathbf{z}$, where randomness is over the $\epsilon$-LDP mechanism $\mathcal{M}$ and the $\alpha$-$d_\sigma$-private shuffling mechanism $\mathcal{A}$. [5]

The adversary wins if it reidentifies $> \frac{k}{2}$ of the LDP values originating from $I$. Let $H = \eta(\mathbf{z})$ be the indices of elements in $\mathbf{z}$ selected by $\eta$. Let $W = \{\sigma \in \mathrm{S}^n : |\sigma(H) \cap I| > \frac{k}{2}\}$ be the set of permutations where the adversary wins and let $L = \{\sigma \in \mathrm{S}^n : \sigma(H) \cap I| \leq \frac{k}{2}\}$ be the set of permutations where the adversary loses.

$$\Pr_{\mathbf{z}}[\eta(\mathbf{z}) \text{ wins}] = \Pr_{\mathbf{z}}[\sigma \in W]$$

$$\Pr_{\mathbf{z}}[\eta(\mathbf{z}) \text{ loses}] = \Pr_{\mathbf{z}}[\sigma \in L]$$

where $\sigma$ is the shuffling permutation produced by $\mathcal{A}$, $\mathbf{z} = \sigma(\mathbf{y})$ i.e. $z_i = y_{\sigma(i)}$. Concretely, this is equivalent to $\mathsf{DO}_i$ releasing $\mathsf{DO}_{\sigma(i)}$'s LDP response. Since the permutation and LDP outputs are randomized, many subgroups of size $k$ in $G_i$ could have produced the LDP values $(z_{H_1}, \ldots, z_{H_k})$ and then been mapped to $H$ by a permutation. Concretely, there is a reasonable probability that Alice's household output the LDP values of another $k$-member household in her neighborhood and they output her household's LDP values. In the worst case, this is $e^{-2k\epsilon}$ less likely than without swapping values, by group DP guarantees. Since both households are part of the same group $G_i$, the permutation that maps her household to elements $H$ in the output is close in probability to that which maps the other household to elements $H$ in the output. As such, we have in the worst case a $e^{-(2k\epsilon+\alpha)}$ reduction in probability of the other household having swapped LDP values with Alice's and permuting to subset $H$.

The above provides intuition on how we could get the same output $\mathbf{z}$ many different ways, and how Alice's household could or could not contribute to elements $H$. It does not, however, explain why an adversary who is given output $\mathbf{z}$ has limited advantage in choosing a subset $H$ such that they recover *most* of Alice's household's values. We formalize this fact as follows.

We may rewrite the probabilities of winning or losing by marginalizing out all possible LDP sequences $\mathbf{y}$. Conditioning on the output sequence $\mathbf{z}$, the only possible LDP sequences $\mathbf{y}$ are permutations of $\mathbf{z}$. Note that the probability of any sequence $\mathbf{y}$ is determined by the input $\mathbf{x}$ and the LDP mechanism $\mathcal{M}$:

$$\Pr_{\mathbf{z}}[\eta(\mathbf{z}) \text{ loses}] = \Pr_{\mathbf{z}}[\sigma \in W]$$

$$= \sum_{\sigma \in W} \Pr\big[\mathcal{A}(\mathbf{x}) = \mathbf{y} = \sigma^{-1}(\mathbf{z})\big] \Pr[\sigma|\mathbf{y}] / \Pr[\mathbf{z}]$$

Note that $\mathrm{Pr}_{\mathbf{z}}[\sigma|\mathbf{y}] = \mathrm{Pr}_{\mathbf{z}}[\sigma]$ for the mallows mechanism, which chooses its permutations independently of $\mathbf{y}$. Now consider when $\eta(\mathbf{z})$ loses. By similar arguments as above:

$$\Pr_{\mathbf{z}}[\eta(\mathbf{z}) \text{ loses}] = \Pr_{\mathbf{z}}[\sigma \in L]$$

$$= \sum_{\sigma \in L} \Pr\big[\mathcal{A}(\mathbf{x}) = \mathbf{y} = \sigma^{-1}(\mathbf{z})\big] \Pr[\sigma|\mathbf{y}] / \Pr[\mathbf{z}]$$

The odds of losing versus winning is given by

$$\frac{\Pr_{\mathbf{z}}[\eta(\mathbf{z}) \text{ loses}]}{\Pr_{\mathbf{z}}[\eta(\mathbf{z}) \text{ wins}]} = \frac{\sum_{\sigma' \in L} \Pr\big[\mathcal{A}(\mathbf{x}) = \mathbf{y} = \sigma'^{-1}(\mathbf{z})\big] \Pr[\sigma'|\mathbf{y}]}{\sum_{\sigma \in W} \Pr[\mathcal{A}(\mathbf{x}) = \mathbf{y} = \sigma^{-1}(\mathbf{z})] \Pr[\sigma|\mathbf{y}]}$$

We now show that for each $\sigma$ in the denominator, we may construct $m = \lfloor \frac{r-k}{k} \rfloor$ distinct permutations $\sigma'$ in the numerator that are close in probability to it.

**Lemma 3.** *For every $\sigma \in W$ there exists a set of $m = \lfloor \frac{r-k}{k} \rfloor$ permutations, $E(\sigma)$, such that*

---

[5] As an abuse of notation, we assume the output space of the LDP randomizers, $\mathcal{Y}$, have outcomes with non-zero measure e.g. randomized response. The following analysis can be expanded to continuous outputs (with outcomes of zero measure) by simply replacing the output sequence $\mathbf{z} \in \mathcal{Y}^n$ with an output event $\mathbf{Z} \subseteq \mathcal{Y}^n$.

1. $E(\sigma) \subseteq L$

2. $\sigma^{-1} \approx_{G_i} \sigma'^{-1}$

3. $E(\sigma_a) \cap E(\sigma_b) = \emptyset$ *for any pair* $\sigma_a, \sigma_b \in W$

4. $\Pr\big[\mathcal{A}(\mathbf{x}) = \mathbf{y} = \sigma^{-1}(\mathbf{z})\big] \leq e^{2k\epsilon} \Pr\big[\mathcal{A}(\mathbf{x}) = \mathbf{y} = \sigma'^{-1}(\mathbf{z})\big]$ *for any* $\mathbf{x} \in \mathcal{X}^n$ *and any* $\mathbf{z} \in \mathcal{Y}^n$

*Proof.* Given $\sigma \in W$, we construct $E(\sigma)$ by first taking the inverse $\sigma^{-1}$. Recall that, since $\sigma \in W$, we have that $|\sigma^{-1}(I) \cap H| > \frac{k}{2}$. ($\sigma^{-1}(i) = j$ could be interpreted as data owner $i$'s LDP value will be output at position $j$). We then divide the remainder of the group $G_i \backslash I$ into $m$ disjoint subsets of size $k$ each, $J_1, J_2, \ldots, J_m$. These represent the other distinct subsets of size $k$ that Alice's household could swap LDP values with. We then produce $m$ permutations, $\sigma_1'^{-1}, \ldots, \sigma_m'^{-1}$, by making $\sigma_i'^{-1}(I) = \sigma^{-1}(J_i)$ and $\sigma_i'^{-1}(J_i) = \sigma^{-1}(I)$ (preserving order within those subsets) and $\sigma'^{-1} = \sigma^{-1}$ everywhere else.

On the first point, we know that every $\sigma' \in E(\sigma)$ is also in $L$. We know this because $\sigma_i'^{-1}(I) = \sigma^{-1}(J_i)$. Since $\sigma \in W$, we have that $|\sigma^{-1}(J_i) \cap H| < \frac{k}{2}$ since $|\sigma^{-1}(I) \cap H| \geq \frac{k}{2}$ and $I \cap J_i = \emptyset$ by definition. Thus, $|\sigma_i'^{-1}(I) \cap H| < \frac{k}{2}$, so $|\sigma_i'(H) \cap I| < \frac{k}{2}$ and $\sigma_i' \in L$.

On the second point, we know that the inverse permutations are neighboring $\sigma^{-1} \approx_{G_i} \sigma'^{-1}$ simply by construction – they only differ on elements in $G_i$.

On the third point, we know that the sets $E(\sigma_a)$ and $E(\sigma_b)$ are distinct since we can map any permutation $\sigma' \in E(\sigma_a)$ uniquely back to $\sigma_a$ for any $\sigma_a \in W$. We do so by taking its inverse $\sigma'^{-1}$, finding which subset $J_i$ has majority elements from $H$ i.e. $|\sigma'^{-1}(J_i) \cap H| > \frac{k}{2}$. Swap elements back: $\sigma'^{-1}(J_i)$ with $\sigma'^{-1}(I)$. Invert back to $\sigma_a$.

On the fourth point, we know that $\sigma^{-1}(\mathbf{z})$ and $\sigma'^{-1}(\mathbf{z})$ differ on at most $2k$ indices. As such, by group DP guarantees, we know that their probabilities must be close to a factor of $e^{-2k\epsilon}$ regardless of $\mathbf{z}$ and $\mathbf{x}$. $\square$

Using the above Lemma we may bound the odds of losing vs. winning.

$$
\begin{aligned}
\frac{\Pr_{\mathbf{z}}[\eta(\mathbf{z}) \text{ loses}]}{\Pr_{\mathbf{z}}[\eta(\mathbf{z}) \text{ wins}]} &= \frac{\sum_{\sigma' \in L} \Pr\big[\mathcal{A}(\mathbf{x}) = \mathbf{y} = \sigma'^{-1}(\mathbf{z})\big] \Pr[\sigma'|\mathbf{y}]}{\sum_{\sigma \in W} \Pr[\mathcal{A}(\mathbf{x}) = \mathbf{y} = \sigma^{-1}(\mathbf{z})] \Pr[\sigma|\mathbf{y}]} \\
&\geq \frac{\sum_{\sigma \in W} \sum_{\sigma' \in E(\sigma)} \Pr\big[\mathcal{A}(\mathbf{x}) = \mathbf{y} = \sigma'^{-1}(\mathbf{z})\big] \Pr[\sigma'|\mathbf{y}]}{\sum_{\sigma \in W} \Pr[\mathcal{A}(\mathbf{x}) = \mathbf{y} = \sigma^{-1}(\mathbf{z})] \Pr[\sigma|\mathbf{y}]} \\
&\geq \min_{\sigma \in W} \frac{\sum_{\sigma' \in E(\sigma)} \Pr\big[\mathcal{A}(\mathbf{x}) = \mathbf{y} = \sigma'^{-1}(\mathbf{z})\big] \Pr[\sigma'|\mathbf{y}]}{\Pr[\mathcal{A}(\mathbf{x}) = \mathbf{y} = \sigma^{-1}(\mathbf{z})] \Pr[\sigma|\mathbf{y}]} \\
&\geq \lfloor \frac{r-k}{k} \rfloor e^{-(2k\epsilon + \alpha)}
\end{aligned}
$$

where the last line follows from the fourth point of the above Lemma (for the $2k\epsilon$ term) and the fact that the inverse permutations $\sigma'^{-1}, \sigma^{-1}$ are neighboring (second point of the Lemma) so the probabilities of the mechanism to produce $\sigma$ vs. $\sigma'$ to reach $\mathbf{z}$ from these neighboring permutations must be close by a factor of $e^{\alpha}$.

Since the above holds for any $\mathbf{z}$ and $\mathbf{x}$, the bound holds on average across all outcomes $\mathbf{z}$, thus

$$\Pr[\eta \text{ loses}] \geq \lfloor \frac{r-k}{k} \rfloor e^{-(2k\epsilon + \alpha)} \cdot \Pr[\eta \text{ wins}]$$

for any deterministic adversary with decision function $\eta$. Finally, we may write any probabilistic adversary as mixture of decision functions. By convexity (same argument used in Lemma 2), the above bound still holds. As such,

$$\Pr[\mathcal{D}_{Adv} \text{ loses}] \geq \lfloor \frac{r-k}{k} \rfloor e^{-(2k\epsilon+\alpha)} \cdot \Pr[\mathcal{D}_{Adv} \text{ wins}]$$

$\square$

### A.7 Utility of Shuffling Mechanism

We now introduce a novel metric, $(\eta, \delta)$-preservation, for assessing the utility of any shuffling mechanism. Let $S \subseteq [n]$ correspond to a set of indices in $\mathbf{y}$. The metric is defined as follows.

**Definition A.4.** $((\eta, \delta)$-preservation) A shuffling mechanism $\mathcal{A} : \mathcal{Y}^n \mapsto \mathcal{Y}^n$ is defined to be $(\eta, \delta)$-preserving $(\eta, \delta \in [0, 1])$ w.r.t to a given subset $S \subseteq [n]$, if

$$\Pr\left[|S_\sigma \cap S| \geq \eta \cdot |S|\right] \geq 1 - \delta, \sigma \in \mathrm{S}_n \tag{9}$$

where $\mathbf{z} = \mathcal{A}(\mathbf{y}) = \sigma(\mathbf{y})$ and $S_\sigma = \{\sigma(i) | i \in S\}$.

For example, consider $S = \{1, 4, 5, 7, 8\}$. If $\mathcal{A}(\cdot)$ permutes the output according to $\sigma = (5\ 3\ 2\ \underline{6}\ \underline{7}\ 9\ \underline{8}\ \underline{1}\ 4\ 10)$, then $S_\sigma = \{5, 6, 7, 8, 1\}$ which preserves 4 or 80% of its original indices. This means that for any data sequence $\mathbf{y}$, at least $\eta$ fraction of its data values corresponding to the subset $S$ overlaps with that of shuffled sequence $\mathbf{z}$ with high probability $(1 - \delta)$. Assuming, $\{y_S\} = \{y_i | i \in S\}$ and $\{z_S\} = \{z_i | i \in S\} = \{y_{\sigma(i)} | i \in S\}$ denotes the set of data values corresponding to $S$ in data sequences $\mathbf{y}$ and $\mathbf{z}$ respectively, we have $\Pr\left[|\{y_S\} \cap \{z_S\}| \geq \eta \cdot |S|\right] \geq 1 - \delta, \forall \mathbf{y}$. For example, let $S$ be the set of individuals from Nevada. Then, for a shuffling mechanism that provides $(\eta = 0.8, \delta = 0.1)$-preservation to $S$, with probability $\geq 0.9, \geq 80\%$ of the values that are reported to be from Nevada in $\mathbf{z}$ are genuinely from Nevada. The rationale behind this metric is that it captures the utility of the learning allowed by $d_\sigma$-privacy – if $S$ is equal to some group $G \in \mathcal{G}$, $(\eta, \delta)$ preservation allows overall statistics of $G$ to be captured. Note that this utility metric is *agnostic of both the data distribution and the analyst's query*. Hence, it is a conservative analysis of utility which serves as a lower bound for learning from $\{z_S\}$. We suspect that with the knowledge of the data distribution and/or the query, a tighter utility analysis is possible.

A formal utility analysis of Alg. 10 is presented in App. A.13. Empirical evaluation of $(\eta, \delta)$ - preservation is presented in App. A.14.

### A.8 Discussion on Properties of Mallows Mechanism

**Property 3.** *For group assignment $\mathcal{G}$, a mechanism $\mathcal{A}(\cdot)$ that shuffles according to a permutation sampled from the Mallows model $\mathbb{P}_{\theta, \mathfrak{d}}(\cdot)$, satisfies $(\alpha, \mathcal{G})$-$d_\sigma$privacy where*

$$\Delta(\sigma_0 : \mathfrak{d}, \mathcal{G}) = \max_{(\sigma, \sigma') \in N_\mathcal{G}} |\mathfrak{d}(\sigma_0 \sigma, \sigma_0) - \mathfrak{d}(\sigma_0 \sigma', \sigma_0)|$$

*and*

$$\alpha = \theta \cdot \Delta(\sigma_0 : \mathfrak{d}, \mathcal{G})$$

*We refer to $\Delta(\sigma_0 : \mathfrak{d}, \mathcal{G})$ as the sensitivity of the rank-distance measure $\mathfrak{d}(\cdot)$*

*Proof.* Consider two permutations of the initial sequence $\mathbf{y}$, $\sigma_1(\mathbf{y}), \sigma_2(\mathbf{y})$ that are neighboring w.r.t. some group $G_i \in \mathcal{G}$, $\sigma_1 \approx_{G_i} \sigma_2$. Additionally consider any fixed released shuffled sequence $\mathbf{z}$. Let $\Sigma_1, \Sigma_2$ be the set of permutations that turn $\sigma_1(\mathbf{y}), \sigma_2(\mathbf{y})$ into $\mathbf{z}$, respectively:

$$\Sigma_1 = \{\sigma \in \mathrm{S}_n : \sigma\sigma_1(\mathbf{y}) = \mathbf{z}\}$$

$$\Sigma_2 = \{\sigma \in \mathrm{S}_n : \sigma\sigma_2(\mathbf{y}) = \mathbf{z}\} \quad .$$

In the case that $\{y\}$ consists entirely of unique values, $\Sigma_1, \Sigma_2$ will each contain exactly one permutation, since only one permutation can map $\sigma_i(\mathbf{y})$ to $\mathbf{z}$.

**Lemma 4.** *For each permutation $\sigma_1' \in \Sigma_1$ there exists a permutation in $\sigma_2' \in \Sigma_2$ such that*

$$\sigma_1' \approx_{G_i} \sigma_2' \quad .$$

Proof follows from the fact that — since only the elements $j \in G_i$ differ in $\sigma_1(\mathbf{y})$ and $\sigma_2(\mathbf{y})$ — only those elements need to differ to achieve the same output permutation. In other words, we may define $\sigma_1', \sigma_2'$ at all inputs $i \notin G_i$ identically, and then define all inputs $i \in G_i$ differently as needed. As such, they are neighboring w.r.t. $G_i$.

Recalling that Alg. 1 applies $\sigma_0^{-1}$ to the sampled permutation, we must sample $\sigma_0\sigma_1'$ (for some $\sigma_1' \in \Sigma_1$) for the mechanism to produce $\mathbf{z}$ from $\sigma_1(\mathbf{y})$. Formally, since $\sigma_1'\sigma_1(\mathbf{y}) = \mathbf{z}$ we must sample $\sigma_0\sigma_1'$ to get $\mathbf{z}$ since we are going to apply $\sigma_0^{-1}$ to the sampled permutation.

$$\Pr\left[\mathcal{A}(\sigma_1(\mathbf{y})) = \mathbf{z}\right] = \mathbb{P}_{\theta,\delta}(\sigma_0\sigma', \sigma' \in \Sigma_1 : \sigma_0)$$

$$\Pr\left[\mathcal{A}(\sigma_2(\mathbf{y})) = \mathbf{z}\right] = \mathbb{P}_{\theta,\delta}(\sigma_0\sigma', \sigma' \in \Sigma_2 : \sigma_0)$$

Taking the odds, we have

$$\frac{\mathbb{P}_{\theta,\delta}(\sigma_0\sigma', \sigma' \in \Sigma_1 : \sigma_0)}{\mathbb{P}_{\theta,\delta}(\sigma_0\sigma'', \sigma'' \in \Sigma_2 : \sigma_0)} = \frac{\sum_{\sigma' \in \Sigma_1} \mathbb{P}_{\Theta,\delta}(\sigma_0\sigma' : \sigma_0)}{\sum_{\sigma'' \in \Sigma_2} \mathbb{P}_{\Theta,\delta}(\sigma_0\sigma'' : \sigma_0)}$$

$$= \frac{\sum_{\sigma' \in \Sigma_1} e^{-\theta\delta(\sigma_0\sigma',\sigma_0)}}{\sum_{\sigma'' \in \Sigma_2} e^{-\theta\delta(\sigma_0\sigma'',\sigma_0)}}$$

$$\leq \frac{e^{-\theta\delta(\sigma_0\sigma_a,\sigma_0)}}{e^{-\theta\delta(\sigma_0\sigma_b,\sigma_0)}}$$

$$\leq e^{\theta|\delta(\sigma_0\sigma_a,\sigma_0)-\delta(\sigma_0\sigma_b,\sigma_0)|}$$

$$\leq e^{\theta\Delta}$$

where

$$\sigma_a = \arg\max_{\sigma' \in \Sigma_1} e^{-\theta\delta(\sigma_0\sigma',\sigma_0)} \text{ and}$$

$$\sigma_a = \arg\min_{\sigma'' \in \Sigma_2} e^{-\theta\delta(\sigma_0\sigma'',\sigma_0)} .$$

Therefore, setting $\alpha = \theta \cdot \Delta$, we achieve $(\alpha, \mathcal{G})$-$d_\sigma$privacy. $\square$

**Property 4.** *The sensitivity of a rank-distance is an increasing function of the width $\omega_\mathcal{G}^{\sigma_0}$. For instance, for Kendall's $\tau$ distance $\delta_\tau(\cdot)$, we have $\Delta(\sigma_0 : \delta_\tau, \mathcal{G}) = \frac{\omega_\mathcal{G}^{\sigma_0}(\omega_\mathcal{G}^{\sigma_0}+1)}{2}$.*

To show the sensitivity of Kendall's $\tau$, we make use of its triangle inequality.

*Proof.* Recall from the proof of the previous property that the expression $\delta(\sigma, \sigma_0) = \delta(\sigma_0\sigma, \sigma_0)$, where $\delta$ is the actual rank distance measure e.g. Kendall's $\tau$. As such, we require that

$$\left|\delta(\sigma_0\sigma_a, \sigma_0) - \delta(\sigma_0\sigma_b, \sigma_0)\right| \leq \frac{\omega_\mathcal{G}^{\sigma_0}(\omega_\mathcal{G}^{\sigma_0} + 1)}{2}$$

for any pair of permutations $(\sigma_a, \sigma_b) \in N_\mathcal{G}$.

For any group $G_i \in \mathcal{G}$, let $W_i \subseteq n$ represent the smallest contiguous subsequence of indices in $\sigma_0$ that contains all of $G_i$.

For instance, if $\sigma_0 = [2, 4, 6, 8, 1, 3, 5, 7]$ and $G_i = \{2, 6, 8\}$, then $W_i = \{2, 4, 6, 8\}$. Then the group width width is $\omega_i = |W_i| - 1 = 3$. Now consider two permutations neighboring w.r.t. $G_i$, $\sigma_a \approx_{G_i} \sigma_b$, so only the elements of $G_i$ are shuffled between them. We want to bound

$$\left|\delta(\sigma_0\sigma_a, \sigma_0) - \delta(\sigma_0\sigma_b, \sigma_0)\right|$$

For this, we use a pair of triangle inequalities:

$$\delta(\sigma_0\sigma_a, \sigma_0\sigma_b) \geq \delta(\sigma_0\sigma_a, \sigma_0) - \delta(\sigma_0\sigma_b, \sigma_0) \quad \& \quad \delta(\sigma_0\sigma_a, \sigma_0\sigma_b) \geq \delta(\sigma_0\sigma_b, \sigma_0) - \delta(\sigma_0\sigma_a, \sigma_0)$$

so,

$$\left|\delta(\sigma_0\sigma_a, \sigma_0) - \delta(\sigma_0\sigma_b, \sigma_0)\right| \leq \delta(\sigma_0\sigma_a, \sigma_0\sigma_b)$$

Since $\sigma_0\sigma_a$ and $\sigma_0\sigma_b$ only differ in the contiguous subset $W_i$, the largest number of discordant pairs between them is given by the maximum Kendall's $\tau$ distance between two permutations of size $\omega_i + 1$:

$$\left|\delta(\sigma_0\sigma_a, \sigma_0\sigma_b)\right| \leq \frac{\omega_i(\omega_i + 1)}{2}$$

Since $\omega_\mathcal{G}^{\sigma_0} \geq \omega_i$ for all $G_i \in \mathcal{G}$, we have that

$$\Delta(\sigma_0 : \delta, \mathcal{G}) \leq \frac{\omega_\mathcal{G}^{\sigma_0}(\omega_\mathcal{G}^{\sigma_0} + 1)}{2}$$

$\square$

## A.9 HARDNESS OF COMPUTING THE OPTIMUM REFERENCE PERMUTATION

**Theorem A.3.** *The problem of finding the optimum reference permutation, i.e., $\sigma_0^* = \arg\min_{\sigma \in S_n} \omega_{\mathcal{G}}^{\sigma}$ is NP-hard.*

*Proof.* We start with the formal representation of the problem as follows.

*Optimum Reference Permutation Problem.* Given n subsets $\mathcal{G} = \{G_i \in 2^{[n]}, i \in [n]\}$, find the permutation $\sigma_0^* = \arg\min_{\sigma \in S_n} \omega_{\mathcal{G}}^{\sigma}$.

Now, consider the following job-shop scheduling problem.

*Job Shop Scheduling.* There is one job $J$ with $n$ operations $o_i, i \in [n]$ and $n$ machines such that $o_i$ needs to run on machine $M_i$. Additionally, each machine has a sequence dependent processing time $p_i$. Let $S$ be the sequence till There are $n$ subsets $S_i \subseteq [n]$, each corresponding to a set of operations that need to occur in contiguous machines, else the processing times incur penalty as follows. Let $p_i$ denote the processing time for the machine running the $i$-th operation scheduled. Let $\mathbb{S}_i$ be the prefix sequence with $i$ schedulings. For instance, if the final scheduling is 1 3 4 5 9 8 10 6 7 2 then $\mathbb{S}_4 = 1345$. Additionally, let $P_{\mathbb{S}_i}^j$ be the shortest subsequence such of $\mathbb{S}_i$ such that it contains all the elements in $S_j \cap \{\mathbb{S}_i\}$. For example for $S_1 = \{3, 5, 7\}$, $P_{\mathbb{S}_4}^1 = 345$.

$$p_i = \max_{i \in [n]}(|P_{\mathbb{S}_i}^j| - |S_j \cap \{\mathbb{S}_i\}|) \tag{10}$$

The objective is to find a scheduling for $J$ such that it minimizes the makespan, i.e., the completion time of the job. Note that $p_n = \max_i p_i$, hence the problem reduces to minimizing $p_n$.

**Lemma 5.** *The aforementioned job shop scheduling problem with sequence-dependent processing time is NP-hard.*

*Proof.* Consider the following instantiation of the sequence-dependent job shop scheduling problem where the processing time is given by $p_i = p_{i-1} + w_{kl}, p_1 = 0$ where $\mathbb{S}_i[i-1] = k$, $\mathbb{S}_i[i] = l$ and $w_{ij}, j \in S_i$ represents some associated weight. This problem is equivalent to the travelling salesman problem (TSP) Balas (2008) and is therefore, NP-hard. Thus, our aforementioned job shop scheduling problem is also clearly NP-hard. □

*Reduction:* Let the $n$ subsets $S_i$ correspond to the groups in $\mathcal{G}$. Clearly, minimizing $\omega_{\mathcal{G}}^{\sigma}$ minimizes $p_n$. Hence, the optimal reference permutation gives the solution to the scheduling problem as well.

□

## A.10 ILLUSTRATION OF ALG. 1

We now provide a small-scale step-by-step example of how Alg. 1 operates.

Fig. 6a is an example of a grouping $\mathcal{G}$ on a dataset of $n = 8$ elements. The group of $\mathsf{DO}_i$ includes $i$ and its neighbors. For instance, $G_8 = \{8, 3, 5\}$. To build a reference permutation, Alg. 1 starts at the index with the largest group, $i = 5$ (highlighted in purple), with $G_5 = \{5, 2, 3, 8, 4\}$. As shown in Figure 6b, the $\sigma_0$ is then constructed by following a BFS traversal from $i = 5$. Each $j \in G_5$ is visited, queuing up the neighbors of each $j \in G_5$ that haven't been visited along the way, and so on. The algorithm completes after the entire graph has been visited.

The goal is to produce a reference permutation in which the width of each group in the reference permutation $\omega_i$ is small. In this case, the width of the largest group $G_5$ is as small as it can be $\omega_5 = 5 - 1 = 4$. However, the width of $G_4 = \{4, 5, 7\}$ is the maximum possible since $\sigma^{-1}(5) = 1$ and $\sigma^{-1}(7) = 8$, so $\omega_4 = 7$. This is difficult to avoid when the maximum group size is large as compared to the full dataset size $n$. Realistically, we expect $n$ to be significantly larger, leading to relatively smaller groups.

With the reference permutation in place, we compute the sensitivity:

$$\Delta(\sigma_0 : \delta, \mathcal{G}) = \frac{\omega_4(\omega_4 + 1)}{2}$$
$$= 28$$

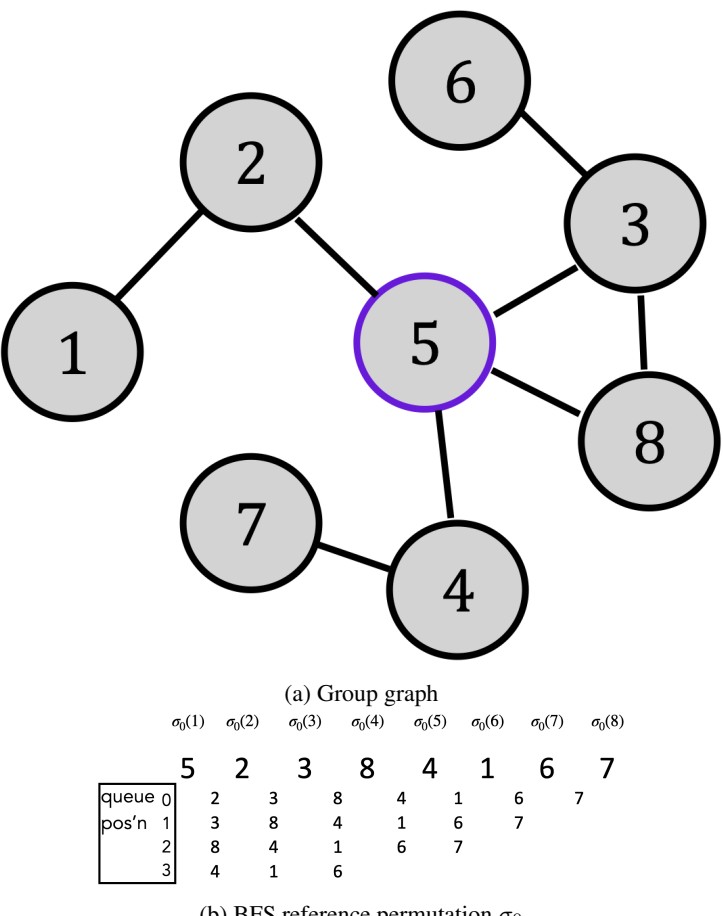

(a) Group graph

| $\sigma_0(1)$ | $\sigma_0(2)$ | $\sigma_0(3)$ | $\sigma_0(4)$ | $\sigma_0(5)$ | $\sigma_0(6)$ | $\sigma_0(7)$ | $\sigma_0(8)$ |
|---|---|---|---|---|---|---|---|
| 5 | 2 | 3 | 8 | 4 | 1 | 6 | 7 |

| | | | | | | | |
|---|---|---|---|---|---|---|---|
| queue 0 | 2 | 3 | 8 | 4 | 1 | 6 | 7 |
| pos'n 1 | 3 | 8 | 4 | 1 | 6 | 7 | |
| 2 | 8 | 4 | 1 | 6 | 7 | | |
| 3 | 4 | 1 | 6 | | | | |

(b) BFS reference permutation $\sigma_0$

Figure 6: Illustration of Alg. 1

Which lets us set $\theta = \frac{\alpha}{28}$ for any given $\alpha$ privacy value. To reiterate, lower $\theta$ results in more randomness in the mechanism.

We then sample the permutation $\hat{\sigma} = \mathbb{P}_{\theta,\boldsymbol{\delta}}(\sigma_0)$. Suppose

$$\hat{\sigma} = [3\,2\,5\,4\,8\,1\,7\,6]$$

Then, the released **z** is given as

$$\mathbf{z} = \sigma^* = \sigma^{-1}\hat{\sigma}(\mathbf{y})$$

$$= [y_1\ y_2\ y_5\ y_8\ y_3\ y_7\ y_6\ y_4]$$

One can think of the above operation as follows. What was 5 in the reference permutation ($\sigma_0(1) = 5$) is 3 in the sampled permutation ($\hat{\sigma}(1) = 3$). So, index 5 corresponding to $\mathsf{DO}_5$ now holds $\mathsf{DO}_3$'s noisy data $y_3$. As such, we shuffle mostly between members of the same group, and minimally between groups.

The runtime of this mechanism is dominated by the Repeated Insertion Model sampler Doignon et al. (2004), which takes $\mathcal{O}(n^2)$ time. It is very possible that there are more efficient samplers available, but RIM is a standard and simple to implement for this first proposed mechanism. Additionally, the majority of this is spent computing sampling parameters which can be stored in advanced with $\mathcal{O}(n^2)$ memory. Furthermore, sampling from a Mallows model with some reference permutation $\sigma_0$ is equivalent to sampling from a Mallows model with the identity permutation and applying it to $\sigma_0$. As such, permutations may be sampled in advanced, and the runtime is dominated by computation of $\sigma_0$ which takes $\mathcal{O}(|V| + |E|)$ time (the number of vertices and edges in the graph).

A future direction could be exploring even better heuristics for computing $\sigma_0$. One possibility could be based on ranked enumeration of the permutations Deep & Koutris (2021); Deep et al. (2021).

## A.11 PROOF OF THM. 4.3

**Theorem 4.3** *Alg. 1 is $(\alpha, \mathcal{G})$-$d_\sigma$ private.*

*Proof.* The proof follows from Prop. 3. Having computed the sensitivity of the reference permutation $\sigma_0$, $\Delta$, and set $\theta = \alpha/\Delta$, we are guaranteed by Property 3 that shuffling according to the permutation $\hat{\sigma}$ guarantees $(\alpha, \mathcal{G})$-$d_\sigma$ privacy.

$\square$

.

## A.12 PROOF OF THM. 4.4

**Theorem 4.4** *Alg. 1 satisfies $(\alpha', \mathcal{G}')$-$d_\sigma$ privacy for any group assignment $\mathcal{G}'$ where $\alpha' = \alpha \frac{\Delta(\sigma_0 : \delta, \mathcal{G}')}{\Delta(\sigma_0 : \delta, \mathcal{G})}$*

*Proof.* Recall from Property 3 that we satisfy $(\alpha, \mathcal{G})$ $d_\sigma$-privacy by setting $\theta = \alpha/\Delta(\sigma_0 : \delta, \mathcal{G})$. Given alternative grouping $\mathcal{G}'$ with sensitivity $\Delta(\sigma_0 : \delta, \mathcal{G}')$, this same mechanism provides

$$
\begin{aligned}
\alpha' &= \frac{\theta}{\Delta(\sigma_0 : \delta, \mathcal{G}')} \\
&= \frac{\alpha/\Delta(\sigma_0 : \delta, \mathcal{G})}{\Delta(\sigma_0 : \delta, \mathcal{G}')} \\
&= \alpha \frac{\Delta(\sigma_0 : \delta, \mathcal{G}')}{\Delta(\sigma_0 : \delta, \mathcal{G})}
\end{aligned}
$$

$\square$

## A.13 FORMAL UTILITY ANALYSIS OF ALG. 1

**Theorem A.4.** *For a given set $S \subset [n]$ and Hamming distance metric, $\delta_H(\cdot)$, Alg. 1 is $(\eta, \delta)$-preserving for $\delta = \frac{1}{\psi(\theta, \delta_H)} \sum_{h=2k+1}^{n} (e^{-\theta \cdot h} \cdot c_h)$ where $k = \lceil (1 - \eta) \cdot |S| \rceil$ and $c_h$ is the number of permutations with hamming distance $h$ from the reference permutation that do not preserve $\eta\%$ of $S$ and is given by*

$$
\begin{aligned}
c_h = \sum_{j=k+1}^{\max(l_s, \lfloor h/2 \rfloor)} \binom{l_s}{j} \cdot \binom{n - l_s}{j} \cdot \Bigg[ \sum_{i=0}^{\min(l_s - j, h - 2j)} \binom{l_s - j}{i} \\
\cdot \binom{i + j}{j} \cdot f(i, j) \cdot \binom{n - l_s - j}{h - 2j - i} \cdot f(h - 2j - i, j)! \Bigg]
\end{aligned}
$$

$$
f(i, 0) = !i, f(0, q) = q!
$$

$$
f(i, j) = \sum_{q=0}^{\min(i,j)} \left[ \binom{i}{q} \cdot \binom{j}{j - q} \cdot j! \cdot f(i - q, q) \right]
$$

$$
l_s = |S|, k = (1 - \eta) \cdot l_s, !n = \lfloor \frac{n!}{e} + \frac{1}{2} \rfloor
$$

*Proof.* Let $l_s = |S|$ denote the size of the set $S$ and $k = \lceil (1 - \eta) \cdot l_S \rceil$ denote the maximum number of correct values that can be missing from $S$. Now, for a given permutation $\sigma \in S_n$, let $h$ denote its Hamming distance from the reference permutation $\sigma_0$, i.e, $h = \delta_H(\sigma, \sigma_0)$. This means that $\sigma$ and $\sigma_0$ differ in $h$ indices. Now, $h$ can be analysed in the the following two cases,

**Case I.** $h \leq 2k + 1$

For $(1 - \eta)$ fraction of indices to be removed from $S$, we need at least $k + 1$ indices from $S$ to be replaced by $k + 1$ values from outside $S$. This is clearly not possible for $h \leq 2k + 1$. Hence, here $c_h = 0$.

**Case II.** $h > 2k$

For the following analysis we consider we treat the permutations as strings (multi-digit numbers are treated as a single string character). Now, Let $\mathbb{S}_{\sigma_0}$ denote the non-contiguous substring of $\sigma_0$ such that it consists of all the elements of $S$, i.e.,

$$|\mathbb{S}| = l_S \tag{11}$$

$$\forall i \in [l_S], \mathbb{S}_{\sigma_0}[i] \in S \tag{12}$$

Let $\mathbb{S}_\sigma$ denote the substring corresponding to the positions occupied by $\mathbb{S}_{\sigma_0}$ in $\sigma$. Formally,

$$|\mathbb{S}_\sigma| = l_S \tag{13}$$

$$\forall i \in [l_S], \mathbb{S}_{\sigma_0}[i] = \sigma(\sigma_0^{-1}(\mathbb{S}_{\sigma_0}[i])) \tag{14}$$

For example, for $\sigma_0 = (1\ 2\ 3\ 5\ 4\ 7\ 8\ 10\ 9\ 6), \sigma = (1\ 3\ 2\ 7\ 8\ 5\ 4\ 6\ 10\ 9)$ and $S = \{2, 4, 5, 8\}$, we have $\mathbb{S}_{\sigma_0} = 2548$ and $S_\sigma = 3784$ where $h = \boldsymbol{\delta}_H(\sigma, \sigma_0) = 9$. Let $\{\mathbb{S}_\sigma\}$ denote the set of the elements of string $\mathbb{S}_\sigma$. Let $A$ be the set of characters in $\mathbb{S}_\sigma$ such that they do not belong to $S$, i.e, $A = \{\mathbb{S}_\sigma[i] | \mathbb{S}_\sigma[i] \notin S, i \in [l_S]\}$. Let $B$ be the set of characters in $\mathbb{S}_\sigma$ that belong to $S$ but differ from $\mathbb{S}_{\sigma_0}$ in position, i.e., $B = \{\mathbb{S}_\sigma[i] | \mathbb{S}_\sigma[i] \in S, \mathbb{S}_\sigma[i] \neq \mathbb{S}_{\sigma_0}[i], i \in [l_S]\}$. Additionally, let $C = S - \{\mathbb{S}_\sigma\}$. For instance, in the above example, $A = \{3, 7\}, B = \{4, 8\}, C = \{2, 5\}$. Now consider an initial arrangement of $p + m$ distinct objects that are subdivided into two types – $p$ objects of Type A and m objects of Type B. Let $f(p, m)$ denote the number of permutations of these $p + m$ objects such that the $m$ Type B objects can occupy any position but no object of Type A can occupy its original position. For example, for $f(p, 0)$ this becomes the number of derangements der denoted as $!p = \lfloor \frac{p!}{e} + \frac{1}{2} \rfloor$. Therefore, $f(|B|, |A|)$ denotes the number of permutations of $\mathbb{S}_\sigma$ such that $\boldsymbol{\delta}_H(\mathbb{S}_{\sigma_0}, \mathbb{S}_\sigma) = |A| + |B|$. This is because if elements of $B$ are allowed to occupy their original position then this will reduce the Hamming distance.

Now, let $\bar{\mathbb{S}}_\sigma$ ($\bar{\mathbb{S}}_{\sigma_0}$) denote the substring left out after extracting from $\mathbb{S}_\sigma$ ($\mathbb{S}_{\sigma_0}$) from $\sigma$ ($\sigma_0$). For example, $\bar{\mathbb{S}}_\sigma = 1256109$ and $\bar{\mathbb{S}}_{\sigma_0} = 1371096$ in the above example. Let $D$ be the set of elements outside of $S$ and $A$ that occupy different positions in $\bar{\mathbb{S}}_\sigma$ and $\bar{\mathbb{S}}_{\sigma_0}$ (thereby contributing to the hamming distance), i.e., $D = \{\bar{\mathbb{S}}_{\sigma_0[i]} | \bar{\mathbb{S}}_{\sigma_0[i]} \notin S, \bar{\mathbb{S}}_{\sigma_0[i]} \neq \bar{\mathbb{S}}_{\sigma[i]}, i \in [n - l_S]\}$. For instance, in the above example $D = \{9, 6, 10\}$. Hence, $h = \boldsymbol{\delta}_H(\sigma, \sigma_0) = |A| + |B| + |C| + |D|$ and clearly $f(|D|, |C|)$ represents the number of permutations of $\bar{\mathbb{S}}_\sigma$ such that $\boldsymbol{\delta}_H(\bar{\mathbb{S}}_\sigma, \bar{\mathbb{S}}_{\sigma_0}) = |C| + |D|$. Finally, we have

$$c_h = \sum_{j=k+1}^{\max(l_s, \lfloor h/2 \rfloor)} \underbrace{\binom{l_s}{j}}_{\text{\# ways of selecting set } C} \cdot \underbrace{\binom{n - l_s}{j}}_{\text{\# ways of selecting set } A} \cdot \Bigg[$$

$$\sum_{i=0}^{\min(l_s - j, h - 2j)} \underbrace{\binom{l_s - j}{i}}_{\text{\# ways of selecting set } B} \cdot f(i, j)$$

$$\cdot \underbrace{\binom{n - l_s - j}{h - 2j - i}}_{\text{\# ways of selecting set } D} \cdot f(h - 2j - i, j) \Bigg]$$

Now, for $f(i, j)$ let $E$ be the set of original positions of Type A that are occupied by Type B objects in the resulting permutation. Additionally, let $F$ be the set of the original positions of Type B objects that are still occupied by some Type B object. Clearly, Type B objects can occupy these $|E| + |F| = m$ in any way they like. However, the type A objects can only result in $f(p - q, q)$ permutations. Therefore, $f(p, m)$ is given by the following recursive function

$$f(p, 0) = !p$$

$$f(0, m) = m!$$

$$f(p, m) = \sum_{q=0}^{\min p, m} \Bigg( \underbrace{\binom{p}{q}}_{\text{\# ways of selecting set } E} \cdot \underbrace{\binom{m}{m - q}}_{\text{\# ways of selecting set } F}$$

$$\cdot m! \cdot f(p - q, q) \Bigg)$$

Thus, the total probability of failure is given by

$$\delta = \frac{1}{\psi(\theta, \mathbf{\delta}_H)} \sum_{h=2k+2}^{n} \left( e^{-\theta \cdot h} \cdot c_h \right) \tag{15}$$

$\square$

### A.14 ADDITIONAL EXPERIMENTAL DETAILS

#### A.14.1 EVALUATION OF $(\eta, \delta)$-PRESERVATION

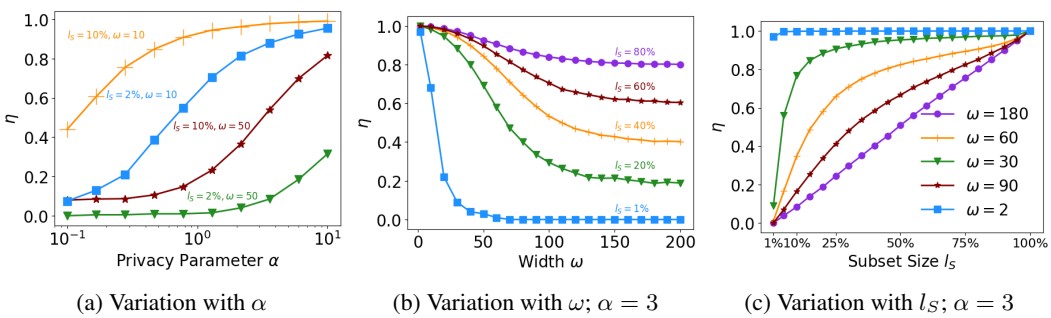

(a) Variation with $\alpha$          (b) Variation with $\omega$; $\alpha = 3$          (c) Variation with $l_S$; $\alpha = 3$

Figure 7: $(\eta, \delta)$-Preservation Analysis

In this section, we evaluate the characteristics of the $(\eta, \delta)$-preservation for Kendall's $\tau$ distance $\mathbf{\delta}_\tau(\cdot, \cdot)$.

Each sweep of Fig. 7 fixes $\delta = 0.01$, and observes $\eta$. We consider a dataset of size $n = 10K$ and a subset $S$ of size $l_S$ corresponding to the indices in the middle of the reference permutation $\sigma_0$ (the actual value of the reference permutation is not significant for measuring preservation). For the rest of the discussion, we denote the width of a permutation by $\omega$ for notational brevity. For each value of the independent axis, we generate 50 trials of the permutation $\sigma$ from a Mallows model with the appropriate $\theta$ (given the $\omega$ and $\alpha$ parameters). We then report the largest $\eta$ (fraction of subset preserved) that at least 99% of trials satisfy.

In Fig. 7a, we see that preservation is highest for higher $\alpha$ and increases gradually with declining width $\omega$ and increasing subset size $l_s$.

Fig. 7b demonstrates that preservation declines with increasing width. $\Delta$ increases quadratically with width $\omega$ for $\mathbf{\delta}_\tau$, resulting in declining $\theta$ and increasing randomness. We also see that larger subset sizes result in a more gradual decline in $\eta$. This is due to the fact that the worst-case preservation (uniform random shuffling) is better for larger subsets. i.e. we cannot do worse than 80% preservation for a subset that is 80% of indices.

Finally, Fig. 7c demonstrates how preservation grows rapidly with increasing subset size. For large widths, we are nearly uniformly randomly permuting, so preservation will equal the size of the subset relative to the dataset size. For smaller widths, we see that preservation offers diminishing returns as we grow subset size past some critical $l_s$. For $\omega = 30$, we see that subset sizes much larger than a quarter of the dataset gain little in preservation.

#### A.14.2 ADULT DATASET

### A.15 ADDITIONAL RELATED WORK

In this section, we discuss the relevant existing work.

The anonymization of noisy responses to improve differential privacy was first proposed by Bittau et al. Bittau et al. (2017a) who proposed a principled system architecture for shuffling. This model was formally studied later in Erlingsson et al. (2019); Cheu et al. (2019). Erlingsson et al. Erlingsson et al. (2019) showed that for arbitrary $\epsilon$-LDP randomizers, random shuffling results in privacy amplification. Cheu et al. Cheu et al. (2019) formally defined the shuffle DP model and analyzed the privacy guarantees of the binary randomized response in this model. The shuffle DP model

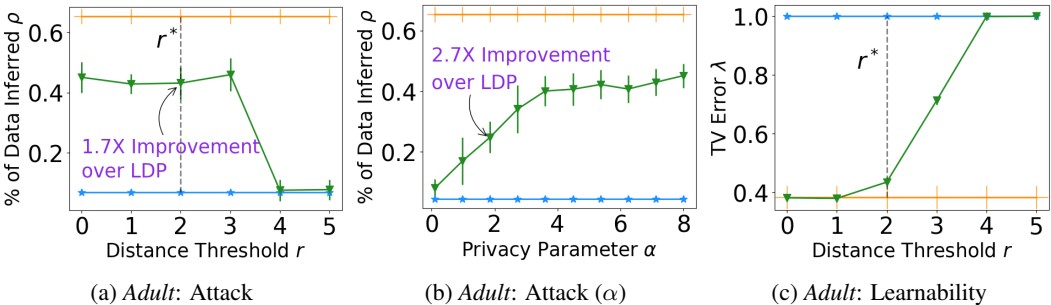

Figure 8: Adult dataset experiments

differs from our approach in two ways. First, it focuses completely on the DP guarantee. The privacy amplification is manifested in the from of a lower $\epsilon$ (roughly a factor of $\sqrt{n}$) when viewed in an alternative DP model known as the central DP model. Erlingsson et al. (2019); Cheu et al. (2019); Balle et al. (2019); Feldman et al. (2020); Bittau et al. (2017a); Balcer & Cheu (2020). However, our result caters to local inferential privacy. Second, the shuffle model involves an uniform random shuffling of the entire dataset. In contrast, our approach the granularity at which the data is shuffled is tunable which delineates a threshold for the learnability of the data.

A steady line of work has sudied the inferential privacy setting Kasiviswanathan & Smith (2014); Kifer & Machanavajjhala (2011); Ghosh & Kleinberg (2016); Dalenius (1977); Dwork & Naor (2010); Tschantz et al. (2020). Kifer et al. Kifer & Machanavajjhala (2011) formally studied privacy degradation in the face of data correlations and later proposed a privacy framework, Pufferfish Kifer & Machanavajjhala (2014); Song et al. (2017); He et al. (2014), for analyzing inferential privacy. Subsequently, several other privacy definitions have also been proposed for the inferential privacy setting Liu et al. (2016); Yang et al. (2015); Chen et al. (2014); Zhu et al. (2015); Bassily et al. (2013). For instance, Gehrke et al. proposed a zero-knowledge privacy Gehrke et al. (2011; 2012) which is based on simulation semantics. Bhaskar et al. proposed noiseless privacy Bhaskar et al. (2011); Grining & Klonowski (2017) by restricting the set of prior distributions that the adversary may have access to. A recent work by Zhang et al. proposes attribute privacy Zhang et al. (2020) which focuses on the sensitive properties of a whole dataset. In another recent work, Ligett et al. study a relaxation of DP that accounts for mechanisms that leak some additional, bounded information about the database Ligett et al. (2020). Some early work in local inferential privacy include profile-based privacy Geumlek & Chaudhuri (2019) by Gehmke et al. where the problem setting comes with a graph of data generating distributions, whose edges encode sensitive pairs of distributions that should be made indistinguishable. In another work by Kawamoto et al., the authors propose distribution privacy Kawamoto & Murakami (2018) – local differential privacy for probability distributions. The major difference between our work and prior research is that we provide local inferential privacy through a new angle – data shuffling.

Finally, older works such as $k$-anonymity Sweeney (2002), $l$-diversity Machanavajjhala et al. (2007), and Anatomy Xiao & Tao (2006) and other Wong et al. (2010); Tassa et al. (2012); Xue et al. (2012); Choromanski et al. (2013); Doka et al. (2015) have studied the privacy risk of non-sensitive auxiliary information, or 'quasi identifiers' (QIs). In practice, these works focus on the setting of dataset release, where we focus on dataset collection. As such, QIs can be manipulated and controlled, whereas we place no restriction on the amount or type of auxiliary information accessible to the adversary, nor do we control it. Additionally, our work offers each individual formal inferential guarantees against informed adversaries, whereas those works do not. We emphasize this last point since formalized guarantees are critical for providing meaningful privacy definitions. As established by Kifer and Lin in *An Axiomatic View of Statistical Privacy and Utility* (2012), privacy definitions ought to at least satisfy post-processing and convexity properties which our formal definition does.

## A.16 EVALUATION OF HEURISTIC

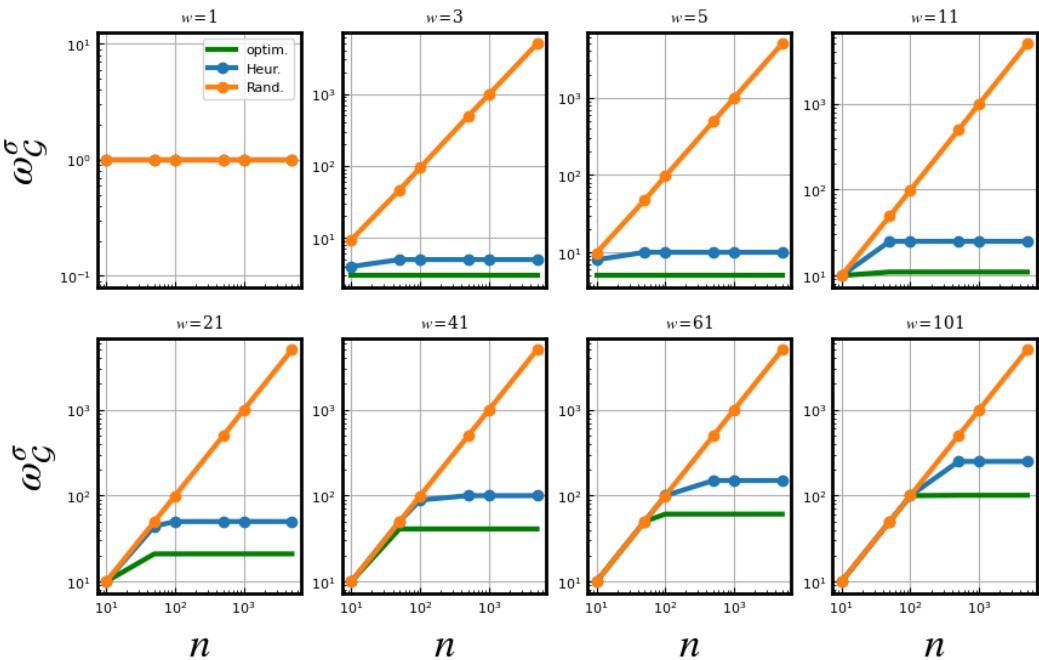

Figure 9: Comparison of our heuristic's performance with that of an optimal reference permutation $\sigma_0^*$. An optimal $\sigma_0^*$ is generated with every group having size $w$. A graph is generated from this optimal $\sigma_0^*$ from which our heuristic (blue) attempts to reconstruct the optimal permutation. For baselining, the performance of a random $\sigma_0$ selection is plotted (orange). We observe that at worst, our heuristic picks a reference permutation with width $2.5\times$ that of the optimal reference permutation (green). See Section 4.4 for definition of terms.

Algorithm 10 is designed to find a reference permutation $\sigma_0$ with low width $\omega_{\mathcal{G}}^{\sigma}$ w.r.t. the given grouping $\mathcal{G}$. A low width is desirable, since it leads to low sensitivity $\Delta(\sigma_0 : \delta, \mathcal{G})$, which in turn leads to higher dispersion parameter $\theta = \alpha/\Delta$, and thus less randomness over permutations (higher utility). Theorem A.3 proves that computing the optimal reference permutation (minimum width) is NP-hard. As such, we propose a BFS-based heuristic.

**Comparison with optimal reference permutation**
To demonstrate the value of the heuristic used in Alg. 10, we provide two evaluations of its performance. For our first evaluation, we compare the performance of our heuristic BFS reference permutation selection ($\sigma_0$) with that of the optimal reference permutation and that of a random reference permutation. As identified by Theorem A.3, finding the optimal reference permutation for a given grouping $\mathcal{G}$ is NP-hard. For these experiments, we first create an optimal reference permutation, where each group $G_i \in \mathcal{G}$ is equally sized $w$ and maximally compact. The optimal width, $\omega_{\mathcal{G}}^{\sigma}$, is then $\min(n, w)$. We then generate a graph from this optimal reference permutation. Finally, we run the BFS reference permutation computation described in Alg. 10 attempting to approximate the optimal $\sigma_0^*$, and compute its width.

To compare with a naive approach, we also plot the performance of a randomly chosen reference permutation. We expect the maximum width across groups $\omega_{\mathcal{G}}^{\sigma}$ to be large for this technique. If one of the $n$ groups has a single entry low (near 0) in $\sigma_0$ and a single entry high (near $n$) in $\sigma_0$, the width will be near $n$. The random baseline is averaged over 10 trials with a 1 standard deviation envelope plotted (but difficult to see, since the variance is low).

Figure 9 depicts our findings. Each plot has a different group size $w$, listed at the top, used in the optimal reference permutation. We find that the random baseline (orange) consistently chooses a reference permutation such that $\omega_{\mathcal{G}}^{\sigma}$ is near $n$, as expected. Our method (blue), on the other hand, closely tracks the optimal solution (green). We find that in the worst case, our algorithm's solution has a width $\leq 2.5\times$ larger than the optimal. Note that for $r = 0$ (upper left), all methods trivially

have a width of one, since the corresponding graph has no edges. While there may be room for improvement, we find this to be sufficient for the present work.

Figure 10: example of our heuristic's performance on randomly generated graphs. As $r$ increases, so does the connectivity of the random graphs and the average group size (green). As shown by Theorem A.3, computing the optimal $\omega_{\mathcal{G}}^{\sigma}$ is NP-hard. The average group size (green) in $\mathcal{G}$ is a loose lower bound on the optimal $\omega_{\mathcal{G}}^{\sigma}$. The performance of a random $\sigma_0$ assignment (orange) is also plotted for reference. Our heuristic BFS algorithm (blue) consistently outperforms the random baseline.

**Performance on randomly generated graphs**

For our second evaluation, we observe how well our BFS heuristic (in Algorithm 10) performs on randomly generated graphs. Here, we sample $n$ points uniformly on the unit interval. We then say that the $i$th point's group, $G_i$, consists of all other points within $r$ of it. As $r$ increases, so does the groups size. Since computing the optimal reference permutation is NP-hard (Theorem A.3), we do not show the optimal width. Instead, we show a loose lower bound of the optimal width (green) by plotting the average group size for a given $r$ (recall that the width is greater than or equal to the largest group size, so we expect this to be a loose lower bound, solely for reference). For comparison, we evaluate the performance of a random $\sigma_0$ choice as well. For both of these methods, we run 10 trials of generating a random graph (and picking a random $\sigma_0$) at each value of $n$ and plot the mean along with a 1 standard deviation envelope, which is difficult to see due to low variance.

Figure 10 depicts our findings. We find that — across values of $n$ and $r$ — our heuristic (blue) significantly outperforms the random baseline (orange). Additionally, we observe the trends we expect. For a low $r$ values, our heuristic BFS algorithm chooses a $\sigma_0$ with width close to the lower bound (green) of the optimal width $\omega_{\mathcal{G}}^{\sigma}$. As $r$ increases, the graph become significantly more connected. Both the lower bound and our heuristic move closer to the width of the random baseline. Note that for $r = 0$ (upper left), all methods trivially have a width of one, since the corresponding graph has no edges. Ultimately, these findings indicate that our heuristic for computing $\sigma_0$ significantly outperforms a naive random choice, and follows the same trend as the lower bound of the optimal.

