# OpenReview forum: "Privacy Implications of Shuffling"
_ICLR.cc/2022/Conference — ICLR 2022 Poster_

### Official Review · Reviewer_ayWA · 2021-11-02

**Correctness:** 3
**Technical Novelty And Significance:** 4
**Empirical Novelty And Significance:** 4
**Recommendation:** 8
**Confidence:** 3

**Details Of Ethics Concerns:**

This is a paper on Differential Privacy, however I don't see any reasons to flag this paper.

**Main Review:**

The paper is original, motivated by allowing more data learnability in the shuffled LDP setting, the authors introduce a new notion of privacy called d_sigma-privacy and show it has desirable properties. Along with the new privacy notion they provide an algorithm that achieves it.
The paper is very well written, clear, and easy to follow, I appreciate the examples. The paper is relevant since it introduces new and useful ideas in the space of privacy and it has practical applications.
I did not verify every single proof but what I looked at seemed correct.

**Summary Of The Paper:**

This paper studies a variant of Local Differential Privacy with shuffling. Local DP is susceptible to inference attacks by using the order of the data. A way to fix the previous is to do “shuffling”, pool LDP responses, apply a permutation and pass that to the analyst. Shuffling gets rid of inference attacks however it severely hurts data learnability. This paper introduces d_sigma-privacy which provides a “tuning knob” between unshuffled LDP and shuffled DP, which rigorously allows trading off privacy and data learnability. Additionally, the authors provide a very general algorithm that achieve’s d_sigma-privacy and run experiments in some datasets validating their results.

**Summary Of The Review:**

The paper is novel and well written. I recommend its acceptance.

---

> ### Author Response · Authors · 2021-11-15
> **Response**
>
> Thank you for your insightful feedback on our paper. We are very gratified to hear that our ideas came across as novel and our privacy guarantees desirable.

---

### Official Review · Reviewer_f7JZ · 2021-11-02

**Correctness:** 3
**Technical Novelty And Significance:** 3
**Empirical Novelty And Significance:** 3
**Recommendation:** 6
**Confidence:** 3

**Details Of Ethics Concerns:**

The paper deals with the problem of privacy preserving learning on data with local neighbourhood structures. An algorithm is proposed and experimentally tested to ensure privacy.

**Main Review:**

Strength:
1. The paper deals with an interesting problem of bridging between local DP and shuffling-based global DP using a neighbouring-information based DP framework. As both the models represent two extremes of assuring DP and also lots of real-life data has a local neighbouring structure, it is pertinent problem to study.
2. The paper proposes to encode the neighbouring information as an undirected graph and use it to define the d_{\sigma} privacy definition. The definition is intuitive and reasonable. It also yields interesting resilience results against Bayesian and decision-theoretic adversaries.
3. A shuffling mechanism is proposed to achieve d_{\sigma} privacy. The mechanism applies Mallow's model while using a heuristic to choose the reference permutation.
4. The experimental evaluations show utility of this new framework and its functionality to bridge between LDP and unif. shuffling DP, which is the central problem of the paper.

Weakness:
1. The heuristics used in the proposed privacy preserving mechanism is not well-analysed and well-justified. Depending on the choice of \sigma_0, the sensitivity can vary exponentially on n or k. Though it is evident that the underlying problem of computing sensitivity is combinatorial and NP-hard, a methodological choice of stable \sigma_0 is expected to ensure stability of the framework. It would be imperative to empirically or theoretically justify the heuristic for choosing \sigma_0 and showing it leads to a stable estimate.
2. How does the privacy level depend on the connectivity of the underlying graph G? For example, how would the privacy level \alpha would vary if the graph changes from a clique to a chain or a completely disconnected graph? An insight on this will be useful to explicitly understand the dependency on existence of neighbouring structure.
3. a. What is (\eta, \delta) preservation? A formal definition would have been helpful. Also, it seems a bit out of place and confusing in the main paper as it never discussed thereafter.
b. though an empirical evaluation is proposed in A.14, it is not clear how to choose a good \eta or \delta, and what are the corresponding trade-offs.
4. The images used in the paper are very hard to read. This part of the paper could probably be edited to have a better presentation. Specially from the beginning to the end, I find a gradual decline in readability and presentation.

**Summary Of The Paper:**

The paper proposes a neighbouring-information based DP framework to bridge between local DP and shuffling-based global DP. The neighbouring information is represented as undirected graph. Depending on this structure, the concept of neighbouring permutations and d_{\sigma} privacy is developed. d_{\sigma} privacy shows interesting resilience results against Bayesian and decision-theoretic adversaries. Further a shuffling mechanism is proposed to achieve this definition. The utility of this framework is validated using experimental evaluations on three datasets.

**Summary Of The Review:**

The paper addresses an interesting problem of assuring DP with a novel framework using neighbouring information of data. The results derived for the decision theoretic and Bayesian attackers are useful and provides insights. A mechanism is proposed to achieve this new DP definition but the method seems to be hastily presented and descriptions of certain concepts are unclear. This is the part where the contribution could be improved. The usefulness of the proposed framework is validated through reasonable experiments.

---

> ### Author Response · Authors · 2021-11-15
> **Response**
>
> Thank you for your insightful feedback on our paper. We provide clarifications for your concerns here:
>
>  1) __Regarding the use of heuristics.__
> Computing sensitivity is *not* NP-hard - given a reference permutation $\sigma_0$, we compute the sensitivity *exactly*. This is necessary for us to uphold our privacy guarantee for a given parameter $\alpha$. Computing the optimal $\sigma_0$ to minimize width $\omega_\mathcal{G}^{\sigma_0}$ (thereby, sensitivity) for a given group assignment, $\mathcal{G}$, is NP-hard (Theorem A.3 in Appendix A.9). However, this does not have any bearing on the privacy guarantee -- it affects only the utility, since lower sensitivity allows for less randomness  in shuffling.
> #
> As an analogy, for standard DP-SGD, the gradients have to be clipped for bounding the sensitivity. Once the clipping threshold $C$ is fixed, the sensitivity of the DP mechanism can be exactly computed. However, the value of $C$ can affect the utility of the algorithm [HCKDP20] and is typically assigned via hyperparameter tuning.  Analogously, we use heuristics to compute $\sigma_0$  for ensuring high utility. We would be happy to provide additional empirical evidence justifying the heuristic in the revised paper.
> #
> 2) __Regarding the connection of $\alpha$ and the underlying graph of the group assignment .__
> First as a semantic comment, the group assignment $\mathcal{G}$, which can be represented by a graph, is an input to our privacy definition along with the privacy parameter $\alpha$ (Definition 4.3 ). In other words, the privacy parameter $\alpha$ is *independent* of $\mathcal{G}$ and our mechanism can achieve $\alpha$-$d_{\sigma}$ privacy $\forall \alpha \in \mathbb{R}_+$ for any arbitrary group assignment $\mathcal{G}$ (equivalently, any graph).  As such, the group assignment $\mathcal{G}$  informs only the *semantic* meaning of the privacy guarantee (i.e., whose noisy releases can be identified and leveraged to make inferences on Alice).
> #
> However, in the case that we design a mechanism based on one group assignment $\mathcal{G}$ and then wish to understand this mechanism’s guarantee on another group assignment $\mathcal{G}’$ (another instantiation of our privacy definition), this is very easy to compute. This exact case is addressed by Theorem. 4.4, which shows that the privacy parameter $\alpha’$ for this new group assignment $\mathcal{G}’$ is equal to the the original privacy parameter, $\alpha$, scaled by the ratio of sensitivities of $\sigma_0$ on $\mathcal{G}$ and $\mathcal{G}’$, i.e.,
> $\alpha'=\alpha\frac{\Delta(\sigma_0 : d, \mathcal{G}')}{\Delta(\sigma_0 : d, \mathcal{G})}$. Since, as discussed above, the sensitivity is very easy to compute from $\sigma_0$, this change in guarantee is also straightforward to compute.
> #
> 3) __Regarding the formal definition of $(\eta, \delta)$ preservation.__
> The formal definition of $(\eta, \delta)$ preservation is given by Definition A.4 in Appendix A.7. Due to lack of space, we present this formal definition and a detailed discussion of $(\eta, \delta)$ preservation in Appendix A.7  which we reference in the main paper shortly after mentioning it (page 7).
> #
> The motivation of $(\eta, \delta)$ preservation is to introduce a formal query-agnostic utility notion for our proposed new shuffling framework. In general, higher the value of $\eta$ (percentage of original indices preserved in a group after shuffling) and lower the value of $\delta$ (probability of failure), better is the utility. Given more space, we would place the A.14 experiments in the main paper along with the definition of ($\eta, \delta$) preservation. However, we left most of the discussion of $(\eta, \delta)$ preservation for the appendix (A.7) since it does not play a role in the experiments of the main paper.  We hope that the discussion in A.7 will help clarify the questions about the additional experiments in Appendix A.14.
>
> 4) __Regarding the presentation of the paper__.
> Due to space constraints, we pack a significant amount of formal content in the final pages. We will revise the presentation of the paper (including figures) to improve the readability.
> #
> [HCKDP20] On the Effectiveness of Mitigating Data Poisoning Attacks with Gradient Shaping

---

> ### Author Response · Authors · 2021-11-22
> **Additional experiments added**
>
> Hi, we have added additional experiments to the appendix in response to your concerns. We would like to reiterate that computing sensitivity is *not* NP-hard; our privacy guarantee is exact. However, we prove that computing the optimal reference permutation $\sigma_0$ to maximize utility is NP-hard, so we propose a heuristic algorithm for choosing the reference permutation.
>
> We have added *new experimental results* for  evaluating our heuristic in Appendix A.16 of the paper. Recall that the quality of a chosen reference permutation is judged by how low its ‘width’ is (the maximum separation between any two points of the same group in the reference permutation). Empirically, we find that the width of the reference permutation chosen by our algorithm stays within 2.5x the width of the optimal choice for large group sizes (up to 101 individuals per group). We additionally find that our heuristic always outperforms a random baseline which uniformly randomly picks the reference permutation.
>
> We are happy to answer any questions regarding these added experiments or other aspects of our work.

---

### Official Review · Reviewer_FmCK · 2021-11-03

**Correctness:** 3
**Technical Novelty And Significance:** 2
**Empirical Novelty And Significance:** 3
**Recommendation:** 6
**Confidence:** 3

**Main Review:**

The model is interesting and strikes a good balance between LDP and Shuffle model. The experimental section also combines utility with plausibility of inference attacks.

The motivation behind the model (even though I think the model is nice) is confusing as the main example is that people in one's household can be used to infer a secret of a particular user. But then the approach proposed by the paper is to reveal that the user is part of some group. Cannot just belonging to this group also be used to infer the privacy parameter? That is, it seems the method just changes which group the user belongs to. It would be good to see why it is ok.

The privacy definition also feel a bit like k-anonymity which is is know to be susceptible to attacks based on belonging to a group. This and above question could probably be answered by providing a more-DP looking definition. That is, can one say you achieve shuffle DP for a group of users? Can one present your definition as a special case of shuffle model and LDP by change \alpha?

The technical method in Alg 1 does not take as input privacy parameters. Hence, it seems that a permutation is chosen only based on G. Can this be changed, i.e., get "more permuted" data is one increases privacy parameters.

**Summary Of The Paper:**

The paper proposes a privacy model in between the guarantees of local DP and shuffle DP. It does so by shuffling each user with a group of "similar" users as opposed to all users. As a result, it achieves better utility than LDP. The group that the user belongs to is publicly available, hence the adversary can tell that the user is one of k particular users but not which one. Hence, the "shuffler" instead of choosing a random uniform  permutation chooses a permutation that satisfies this per-group shuffle. It is intractable to choose a permutation with these properties, hence the authors propose a heuristic method for choosing such permutation.

**Summary Of The Review:**

A very well written paper and nice experimental section.
The paper can be strengthened with motivation of the new definition and its (formal) relation to existing definitions.

---

> ### Author Response · Authors · 2021-11-15
> **Response**
>
> Thank you for your insightful feedback on our paper. We provide clarifications to your concerns here.
>
> 1) __Regarding the knowledge of the group a user belongs to.__
> The knowledge that a given user belongs to a given group is *public* and known apriori (owing to public auxiliary information about user - see Section 1 and Section 4.2) -- our mechanism does *not* reveal this knowledge. For instance, in the example used in the introduction (Section 1),  the fact that Alice belongs to a particular household/neighborhood/township is known apriori to the adversary from her home address which is a public auxiliary information. Our privacy definition, $d_{\sigma}$, is motivated to provide protection against inference attacks that can leverage this public knowledge.
> #
> Using LDP alone, the noisy responses can be linked with individuals and lead to inference attacks. For instance, using the public auxiliary information of home address, the noisy reponses of Alice’s household can then be identified and used to infer her true disease state. With $d_\sigma$ privacy, this public knowledge is *no longer a threat*.
> Specifically, $d_{\sigma}$ inputs a group assignment $G_i$ for every user $i$ which determines their *choice of privacy/utility trade-off*. In our example, if Alice chooses her group $G_{Alice}$ for the $d_{\sigma}$ definition  to be that of her neighborhood then, the noisy responses from her household cannot be re-identified within the shuffled release. Thus, Alice’s household’s responses *cannot* be specifically leveraged to make inferences about her (privacy). We make this mathematically precise with our two semantic guarantees (Theorem 4.1 and Theorem 4.2).  Meanwhile, the order is preserved well enough in the shuffled release to allow the analyst to study a broader trend (utility) -- disease prevalence in her neighborhood.
>
> #
> #
>
> 2) __Regarding connections to LDP, shuffle DP and k-anonymity.__
> Our privacy guarantee $d_{\sigma}$ (Definition 4.3) is in fact a generalization of the shuffle model. If $\alpha=0$ then, $d_{\sigma}$ boils down to the standard shuffle DP (uniform random shuffling). If $\alpha=\infty$ then, $d_{\sigma}$ is equivalent to the standard LDP (no shuffling). Thus, $d_{\sigma}$ interpolates between the two extremes and allows us to explore the rich class of privacy/utility trade-offs represented by all the intermediate points in terms of protection against inference attacks.
> #
> While there are some notional similarities between our definition and k-anonymity, there are critical differences. We discuss this in some detail in Appendix A.15. First, k-anonymity offers no formal privacy guarantee, whereas we provide formal privacy guarantees (Definition 4.3, Theorem 4.1, Theorem 4.2). Second, k-anonymity is not intended to be used with LDP, where our definition is. The privacy offered by the combination of our mechanism with LDP is identified by Theorem 4.2. Third, k-anonymity only cares about the size of the group, not who is in the group -- a significant difference with our definition that allows us to prevent inference attacks using specific individuals (for instance, Alice’s household). Finally, k-anonymity does not provide for overlapping groups -- a non-trivial problem, which we solve.
> #
> The primary attack in the literature on k-anonymity type methods is the De Finetti attack. We provide a formal analysis in Appendix A3 of how a strict instance of $d_\sigma$ privacy is necessary and sufficient to thwart such an attack.
> #
>
> 3) __Regarding the privacy parameter for Alg. 1.__
> The presentation of Alg. 1 on page 7 does take the privacy parameter $\alpha$ as an input (third line under “Inputs” in Alg. 1). $\alpha$ is then used to compute the dispersion parameter $\theta$ (line 6). We will make this more visible in the revised version. Your intuition is correct: as $\alpha$ decreases, data is ‘more permuted’ and the privacy is enhanced. We demonstrate this empirically in Figure 8b of the Appendix (page 28).

---

> ### Author Response · Authors · 2021-11-22
> **Check-in for further clarifications and comments**
>
> We would like to check in to see if we can provide any further clarifications regarding our work. In our previous response we 1) reiterate the motivation of our work and clarify how the knowledge of belonging to a certain group is public and *not* a disclosure from our scheme 2) show how our work is a generalization of the shuffle model, with shuffle DP and LDP being its two extremes, and is distinguished from prior works like k-anonymity and 3) clarify that our algorithm does indeed use the privacy parameter as input, and its relation to utility.

---

### Decision · Program_Chairs · 2022-01-20

**Decision:**

Accept (Poster)

**Comment:**

Thank you for your submission. The reviewers agree that this paper provides new contributions to data privacy. In particular, the proposed definition interpolates between the local differential privacy and shuffled differential privacy definitions. As argued in the paper, mechanisms under this framework can prevent certain inferential attacks based on the relationships across the individuals (e.g., which individuals belong to the same household). The paper also provides good evidence that their mechanism guards against a specific type of inferential attacks and provides stronger utility than mechanisms based on uniform shuffling.